# Quantum spin systems versus Schrödinger operators: A case study in spontaneous symmetry breaking

**Christiaan J. F. van de Ven**[1][⋆], **Gerrit C. Groenenboom**[2],
**Robin Reuvers**[3] **and Nicolaas P. Landsman**[4]

**1** Department of Mathematics, University of Trento, INFN-TIFPA, Trento, Italy. Marie
Skłodowska-Curie Fellow of the Istituto Nazionale di Alta Matematica.
**2** Theoretical Chemistry, Institute for Molecules and Materials,
Radboud University, Nijmegen, The Netherlands.
**3** Department of Applied Mathematical and Theoretical Physics,
University of Cambridge, Cambridge, U.K.
**4** Institute for Mathematics, Astrophysics, and Particle Physics,
Radboud University, Nijmegen, The Netherlands.

⋆ christiaan.vandeven@unitn.it

## Abstract

Spontaneous symmetry breaking (SSB) is mathematically tied to some limit, but must physically occur, approximately, *before* the limit. Approximate SSB has been independently understood for Schrödinger operators with double well potential in the *classical limit* [1, 2] and for quantum spin systems in the *thermodynamic* limit [3, 4]. We relate these to each other in the context of the Curie–Weiss model, establishing a remarkable relationship between this model (for finite $N$) and a discretized Schrödinger operator with double well potential.

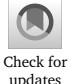

## 1 Introduction

At first sight, spontaneous symmetry breaking (SSB) is a paradoxical phenomenon: in *Nature*, finite quantum systems, such as crystals, evidently display it, yet in *Theory* it seems forbidden in such systems. Indeed, for finite quantum systems the ground state of a generic Hamiltonian is unique and hence invariant under whatever symmetry group $G$ it may have.[1] Hence SSB, in the sense of having a family of asymmetric ground states related by the action of $G$, seems possible only in infinite quantum systems or in classical systems (for both of which the arguments proving uniqueness, typically based on the Perron–Frobenius Theorem break down, cf. Appendix A). However, both are idealizations, vulnerable to what we call *Earman's Principle* from the philosophy of physics:

> "While idealizations are useful and, perhaps, even essential to progress in physics,
> a sound principle of interpretation would seem to be that no effect can be counted
> as a genuine physical effect if it disappears when the idealizations are removed."
> [5]

As argued in detail in Ref. [6–8], the solution to his paradox lies in Earman's very principle itself, which (contrapositively) implies what we call *Butterfield's Principle*:

> "there is a weaker, yet still vivid, novel and robust behaviour that occurs before we
> get to the limit, i.e. for finite $N$. And it is this weaker behaviour which is physically
> real." [9]

Applied to SSB in infinite quantum systems, this means that some approximate and robust form of symmetry breaking should already occur in large but finite systems, *despite the fact that uniqueness of the ground state seems to forbid this*. Similarly, SSB in a classical system should be foreshadowed in the quantum system whose classical limit it is, at least for tiny but positive values of Planck's constant $\hbar$. To accomplish this, it must be shown that for finite $N$ or $\hbar > 0$ the system is not in its ground state, but in some other state having the property that as $N \rightarrow \infty$ or $\hbar \rightarrow 0$, it converges in a suitable sense (detailed in Ref. [7], Chapter 8 and Chapter 7, respectively) to a symmetry-broken ground state of the limit system, which is either an infinite quantum system or a classical system. Since the symmetry of a state is preserved under the limits in question (provided these are taken correctly), this implies that *the actual physical state at finite $N$ or $\hbar > 0$ must already break the symmetry*. The mechanism to accomplish this, originating with Anderson [3], is based on forming symmetry-breaking linear combinations of low-lying states (sometimes called "Anderson's tower of states") whose energy difference vanishes in the pertinent limit.[2] This mechanism has been independently

---

[1]Similarly for equilibrium states at positive temperature, which are always (not just generically) unique.

[2]It must be admitted that the description in Anderson [3] and even in his textbook [10] is very brief and purely qualitative, and that in Ref. [10], which he calls "my most complete summary of the theory of broken symmetry in condensed matter systems" the idea is not even mentioned.

verified in two different contexts, namely quantum spin systems in the thermodynamic limit,[3] and $1d$ Schrödinger operators with a symmetric double well potential in the classical limit.[4] In the absence of cross-references so far, one of our contributions will be to map the specific perturbations that play a key role in SSB for Schrödinger operators at $\hbar > 0$ onto quantum spin systems.

To this end, we now briefly recall the main point of Jona-Lasinio [1], later called the "flea on the elephant" [2] or, in applications to the measurement problem, the "flea on Schrödinger's Cat" [8]. Consider the Schrödinger operator with symmetric double well, defined on suitable domain in $\mathcal{H} = L^2(\mathbb{R})$ by

$$h_\hbar = -\hbar^2 \frac{d^2}{dx^2} + \tfrac{1}{4}\lambda(x^2 - a^2)^2, \tag{1.1}$$

where $\lambda > 0$ and $a \neq 0$. For any $\hbar > 0$ the ground state of this Hamiltonian is unique and hence invariant under the $\mathbb{Z}_2$-symmetry $\psi(x) \mapsto \psi(-x)$; with an appropriate phase choice it is real, strictly positive, and doubly peaked above $x = \pm a$. Yet the classical Hamiltonian

$$h_0(p,q) = p^2 + \tfrac{1}{4}\lambda(q^2 - a^2)^2, \tag{1.2}$$

defined on the classical phase space $\mathbb{R}^2$, has a two-fold degenerate ground state: the point(s) $(p_0, q_0)$ in $\mathbb{R}^2$ where $h_0$ takes an absolute minimum are $(p_0 = 0, q_0 = \pm a)$. In the algebraic formalism, where states are defined as normalized positive linear functionals on the C*-algebra $A_0 = C_0(\mathbb{R}^2)$, the (pure) ground states are the *asymmetric* Dirac measures

$$\omega_\pm(f) = f(p = 0, q = \pm a). \tag{1.3}$$

From these, one may construct the mixed *symmetric* state

$$\omega_0 = \tfrac{1}{2}(\omega_+ + \omega_-), \tag{1.4}$$

which in fact is the limit of the (C*-algebraic) ground state $\omega_\hbar$ of (1.1) as $\hbar \to 0$, where

$$\omega_\hbar(a) = \langle \psi_\hbar^{(0)}, a\psi_\hbar^{(0)}\rangle, \tag{1.5}$$

in terms of the usual ground state $\psi_\hbar^{(0)} \in L^2(\mathbb{R})$ of $h_\hbar$ (assumed to be a unit vector).[5] In order to have a quantum "ground-ish" state that converges to either one of the physical classical ground states $\omega_+$ or $\omega_-$ rather than to the unphysical mixture $\omega_0$, we perturb (1.1) by adding an asymmetric term $\delta V$ (i.e., the "flea"), which, however small it is, under reasonable assumptions localizes the ground state $\psi_\hbar^{(\delta)}$ of the perturbed Hamiltonian in such a way that $\omega_\hbar^{(\delta)} \to \omega_+$ or $\omega_-$, depending on the sign and location of $\delta V$.[6] In particular, the localization of $\psi_\hbar^{(\delta)}$ grows exponentially as $\hbar \to 0$ (see §4 for details). In this paper, we adapt this scenario to the *Curie–Weiss model* on a finite lattice $\Lambda \subset \mathbb{Z}^d$, with Hamiltonian

$$h_\Lambda^{\text{CW}} = -\tfrac{1}{2}|\Lambda|^{-1} \sum_{x,y\in\Lambda} \sigma_3(x)\sigma_3(y) - B \sum_{x\in\Lambda} \sigma_1(x). \tag{1.6}$$

---

[3]See e.g. the reviews [11–14], as well as the original papers [15–25], and, rigorously, [4].

[4]Three founding papers are [1,2,26]. Since in the context of Schrödinger operators the classical limit "$\hbar \to 0$" typically means that $m \to \infty$ at fixed $\hbar$ (where $m$ is the mass occurring in $\hbar^2/2m$), one may physically see $\hbar \to 0$ as a special case of $N \to \infty$.

[5] [7], §7.1. Here $\omega_\hbar$ is defined as a normalized positive linear functional on the C*-algebra $A_\hbar = B_0(L^2(\mathbb{R}))$ of compact operators on $L^2(\mathbb{R})$. The algebraic formalism is particularly useful for combining classical and quantum expressions.

[6] For example, if $\delta V$ is positive and is localized to the right, then the relative energy in the left-hand part of the double well is lowered, so that localization will be to the left. See §4 for details.

Here we take the spin-spin coupling to be $J = 1$, and $B$ is an external magnetic field.[7]
This Hamiltonian has a $\mathbb{Z}_2$-symmetry $(\sigma_1, \sigma_2, \sigma_3) \mapsto (\sigma_1, -\sigma_2, -\sigma_3)$, which at each site $x$ is
implemented by $u(x) = \sigma_1(x)$. The ground state of this model is unique for any $|\Lambda| < \infty$
and any $B \neq 0$, and yet, as for the double well potential, in the thermodynamic limit it has
two degenerate ground states, provided $0 < B < 1$. As explained in Ref. [7, 29–31], perhaps
unexpectedly this limit actually defines a *classical* theory, with phase space $B^3 \subset \mathbb{R}^3$, i.e. the
three-sphere with unit radius (and boundary $\partial B^3 = S^2$), seen as a Poisson manifold with
bracket $\{x, y\} = z$ etc.), and Hamiltonian

$$h_0^{\text{CW}}(x, y, z) = -\tfrac{1}{2}z^2 - Bx. \tag{1.7}$$

The ground states of this Hamiltonian are simply its absolute minima, viz. ($\vec{x} = (x, y, z)$):

$$\vec{x}_\pm = (B, 0, \pm\sqrt{1 - B^2}) \ (0 \leq B < 1); \tag{1.8}$$
$$\vec{x} = (1, 0, 0) \ (B \geq 1), \tag{1.9}$$

which lie on the boundary $S^2$ of $B^3$ (note that the points $\vec{x}_\pm$ coalesce as $B \to 1$, where they
form a saddle point). Thus we seem to face a similar paradox as for the double well.
To address this, in §2 we first show that due to permutation invariance and strict positivity
of the ground state of the 1$d$ Curie–Weiss Hamiltonian (for $N < \infty$ and $|\Lambda| = N$), which is
initially defined on the $2^N$-dimensional Hilbert space $\mathcal{H}_N = \bigotimes_{n=1}^N \mathbb{C}^2$, the ground state of this
Hamiltonian must lie in the range $\text{ran}(S_N)$ of the appropriate symmetriser

$$S_N(v) = \frac{1}{N!} \sum_{\sigma \in S_n} L_\sigma(v) \tag{1.10}$$

on $\mathcal{H}_N$; here $v$ is a vector in the $N$-fold tensor product and $L_\sigma$ is given by permuting the factors
of $v$, i.e. $v_1 \otimes \cdots \otimes v_n \mapsto v_{\sigma(1)} \otimes \cdots \otimes v_{\sigma(n)}$. Its range is $(N+1)$-dimensional, and we show that the
quantum Curie–Weiss Hamiltonian restricted to $\text{ran}(S_N)$ becomes a *tridiagonal* $(N+1) \times (N+1)$
matrix. Even for large $N$, this matrix is easy to diagonalize numerically. Using this tridiagonal
structure, in §3 we show that as $N \to \infty$, our restricted Curie–Weiss Hamiltonian (rescaled
by a factor $1/N$) increasingly well approximates a 1$d$ Schrödinger operator with a symmetric
double well potential defined on the interval $[0, 1]$, in which $\hbar = 1/N$. In §4 we use these ideas
to find the counterpart of the "flea" perturbation (symmetry breaking field) from the double
well potential for the Curie–Weiss model, which, analogously to the double well, localizes the
ground state of the perturbed Hamiltonian (this time, of course, in spin configuration space
rather than real space).

Although our physical mechanism for SSB in finite systems is the same as the one studied
in the condensed matter physics literature (namely Anderson's), the mathematical form of
the flea perturbation is a bit different from the usual symmetry-breaking terms for quantum
spin systems, which in the double well would correspond to breaking the symmetry by simply
deepening one of the bottoms or changing its curvature, whereas the "flea" is typically localized
away from the minima, cf. [26]. In the mathematical framework in which we work, our
approach also has the advantage of making the limits $\hbar \to 0$ and $N \to \infty$ quite regular (i.e.,
continuous, properly understood), as opposed to the alternative view of regarding them as
singular (e.g. [13, 27, 28]). This is further discussed in our Conclusion, which also states some

---

[7] Note that putting $J = 1$ makes $h_\Lambda^{\text{CW}}$ dimensionless. This model falls into the class of homogeneous mean-field theories, see [29–31], which differ from their short-range counterparts (which in this case would be the quantum Ising model) in that every spin now interacts with every other spin. This also makes the dimension $d$ irrelevant (which marks a huge difference with short-range quantum spin models), and yet even the apparently simple Curie–Weiss model is extremely rich in its behaviour; see e.g. [32] for a detailed analysis (along quite different lines from our study), motivated by the measurement problem.

113 open problems and suggestions for further research. This is followed by an appendix on the
114 Perron–Frobenius Theorem, which plays a central role in our work, and another appendix
115 introducing the discretization techniques we use to non-specialists.

# 2 Reduction of the Curie–Weiss Hamiltonian

117 Since the spatial dimension is irrelevant, we may as well consider the Curie–Weiss Hamiltonian
118 (1.6) in $d = 1$, so that we may simply write $|\Lambda| = N$, and, with $h_N \equiv h_N^{CW}$,

$$h_N = -\tfrac{1}{2}N^{-1} \sum_{x,y=1}^{N} \sigma_3(x)\sigma_3(y) - B \sum_{x=1}^{N} \sigma_1(x). \tag{2.1}$$

119 It seems folklore that the Perron-Frobenius theorem yields uniqueness and strict positivity of
120 the ground state $\psi_N^{(0)}$ of $h_N$ for any $N < \infty$ and $B \neq 0$, but for completeness we provide the
121 details in Appendix A. It follows that $\psi_N^{(0)}$ is $\mathbb{Z}_2$-invariant (see the Introduction), so that on a
122 first analysis (to be corrected in what follows!) there is no SSB for any finite $N$.

## 2.1 Tridiagonal form

124 Let $S_N$ be the standard symmetriser (1.10) on the Hilbert space $\mathcal{H}_N = \bigotimes_{n=1}^{N} \mathbb{C}^2$ on
125 which $h_N$ acts, so that $S_N$ projects onto the subspace $\mathrm{ran}(S_N) = \mathrm{Sym}^N(\mathbb{C}^2)$ of totally
126 symmetrised tensors. An orthonormal basis for $\mathrm{Sym}^N(\mathbb{C}^2)$ is given by the vectors
127 $\{|n_+, n_-\rangle \mid n_+ = 0, ..., N, \ n_+ + n_- = N\}$, where $|n_+, n_-\rangle$ is the totally symmetrised unit vector
128 in $\bigotimes_{n=1}^{N} \mathbb{C}^2$, with $n_+$ spins up and $n_- = N - n_+$ spins down. It follows that this space is
129 $(N + 1)$-dimensional. Since $h_N$ commutes with all permutations of $\{1, ..., N\}$, in view of its
130 uniqueness the ground state $\psi_N^{(0)}$ of $h_N$ must be invariant under the permutation group and
131 hence under $S_N$. Hence we may expand $\psi_N^{(0)}$ according to

$$\psi_N^{(0)} = \sum_{n_+=0}^{N} c(n_+/N)|n_+, n_-\rangle, \tag{2.2}$$

132 where we conveniently introduce a function $c : \{0, 1/N, 2/N, ..., (N-1)/N, 1\} \to [0, 1]$ that
133 satisfies $c(n_+/N) = c(n_-/N)$ as well as $\sum_{n_+=0}^{N} c^2(n_+/N) = 1$.

134 **Theorem 2.1.** *In the basis* $\{|n_+\rangle\} \equiv \{|n_+, N - n_+\rangle\}$, *the operator* (2.1) *is an* $(N + 1) \times (N + 1)$
135 *tridiagonal matrix:*[8]

$$-\frac{1}{2N}(2n_+ - N)^2 \ \text{ on the diagonal;} \tag{2.3}$$

$$-B\sqrt{(N - n_+)(n_+ + 1)} \ \text{ on the upper diagonal;} \tag{2.4}$$

$$-B\sqrt{(N - n_+ + 1)n_+} \ \text{ on the lower diagonal.} \tag{2.5}$$

136 *Proof.* Given two arbitrary vectors $|n_+\rangle$ and $|n_+'\rangle$, we have to compute the expression

$$\langle n_+|h_N^{CW}|n_+'\rangle, \quad (n_+, n_+' = 0, ..., N); \tag{2.6}$$

---

[8]The following relations can also be derived from those in Appendix 2 of Ref. [33].

where we we have used the bra-ket notation in the above expression. By linearity, we may separately compute this for the operators

$$
h_N^{(1)} = \sum_{x,y=1}^{N} \sigma_3(x)\sigma_3(y) = \sum_{x=1}^{N} \sigma_3(x) \cdot \sum_{y=1}^{N} \sigma_3(y),
$$
$$
h_N^{(2)} = \sum_{x=1}^{N} \sigma_1(x). \tag{2.7}
$$

In order to compute (2.6), we need to know how $\sigma_3$ and $\sigma_1$ act on the vectors $|n_+\rangle$. Consider the standard basis $\{e_1, e_2\}$ for $\mathbb{C}^2$ over $\mathbb{C}$. Then $\{e_{n_1} \otimes ... \otimes e_{n_N}\}_{n_1=1,...,n_N=1}^{2}$ is the standard basis for $\bigotimes_{n=1}^{N} \mathbb{C}^2$. Note that $\sigma_3(x) = 1 \otimes ... \otimes 1 \otimes \sigma_3 \otimes 1... \otimes 1$, where $\sigma_3$ acts on the $x^{th}$ position and similarly for $\sigma_1(x)$. It follows that for all $x, y \in \{1, ..., N\}$,

$$
\sigma_3(x)(e_{n_1} \otimes ... \otimes e_{n_N}) =
$$
$$
1(e_{n_1}) \otimes ... \otimes 1(e_{n_{x-1}}) \otimes \sigma_3(e_{n_x}) \otimes 1(e_{n_{x+1}}) \otimes ... \otimes 1(e_{n_N}) =
$$
$$
\begin{cases} +(e_{n_1} \otimes ... \otimes e_{n_N}), & \text{if } e_{n_x} = \begin{pmatrix} 1 \\ 0 \end{pmatrix} \\ -(e_{n_1} \otimes ... \otimes e_{n_N}), & \text{if } e_{n_x} = \begin{pmatrix} 0 \\ 1 \end{pmatrix}. \end{cases} \tag{2.8}
$$

We have $\sigma_3(y)\sigma_3(x)(e_{n_1} \otimes ... \otimes e_{n_N}) = \pm(e_{n_1} \otimes ... \otimes e_{n_N})$, where the minus sign appears only if $e_{n_x} \neq e_{n_y}$. We conclude that the standard basis for the $N$-fold tensor product is a set of eigenvectors for $\sigma_3(y)\sigma_3(x)$ with eigenvalues $\pm 1$. Thus we know that $\sum_{x,y} \sigma_3(x)\sigma_3(y)$ is a diagonal matrix with respect to this standard basis. Note that $|n_+\rangle$ is a (normalized) sum of permutations of such basis vectors, with $n_+$ times the vector $e_1$ and $N - n_+$ times the vector $e_2$. Since $\sum_{x,y} \sigma_3(y)\sigma_3(x)$ acts diagonally on any of these vectors, and it is also permutation invariant, it follows that in the inner product any vector with itself yields the same contribution, namely

$$
\langle e_1 \otimes \cdots \otimes e_1 \otimes e_2 \cdots \otimes e_2 | \sum_{x,y} \sigma_3(y)\sigma_3(x) | e_1 \otimes \cdots \otimes e_1 \otimes e_2 \cdots \otimes e_2 \rangle, \tag{2.9}
$$

where $e_1$ occurs $n_+$ times and $e_2$ occurs $N - n_+$ times. The above expression equals

$$
(n_+ - (N - n_+))^2 = (2n_+ - N)^2,
$$

since there are $2n_+(N - n_+)$ minus signs and hence $N^2 - 2n_+(N - n_+) = n_+^2 + (N - n_+)^2$ plus signs, so that in total the correct value is indeed given by

$$
n_+^2 + (N - n_+)^2 - 2n_+(N - n_+) = (2n_+ - N)^2. \tag{2.10}
$$

This shows that the contribution to the diagonal is given by (2.3).

In order to compute the off-diagonal contribution (2.6) with (2.7), we use an explicit expression for the symmetric basis vector $|n_+\rangle$. Using (1.10), it is easy to show that

$$
|n_+\rangle = \frac{1}{\sqrt{\binom{N}{n_+}}} \sum_{l=1}^{\binom{N}{n_+}} \beta_{n_+,l}, \tag{2.11}
$$

where the subindex $l$ in $\beta_{n_+,l}$ labels the possible permutations of the factors in the basis vector $\beta_{n_+,l}$. Since there are $\binom{N}{n_+}$ such permutations, the subindex indeed goes from 1 to $\binom{N}{n_+}$. We fix $N$ and $n_+$, and put

$$W^1_{n_+} = \{y \in \{1,...,N\} | \; \beta_{n_+} \text{ has } e_1 \text{ on position } y \}, \quad \text{and}$$
$$W^2_{n_+} = \{y \in \{1,...,N\} | \; \beta_{n_+} \text{ has } e_2 \text{ on position } y \}. \tag{2.12}$$

Then

$$\#W^1_{n_+} + \#W^2_{n_+} = n_+ + (N - n_+) = n_+ + n_- = N. \tag{2.13}$$

Both sets are clearly disjoint. Then we compute

$$\frac{1}{\sqrt{\binom{N}{n_+}}} \frac{1}{\sqrt{\binom{N}{n'_+}}} \sum_{l=1}^{\binom{N}{n_+}} \sum_{k=1}^{\binom{N}{n'_+}} \langle \beta_{n_+,l} | h_N^{(2)} | \beta_{n'_+,k} \rangle = \frac{1}{\sqrt{\binom{N}{n_+}}} \frac{1}{\sqrt{\binom{N}{n'_+}}} \sum_{l=1}^{\binom{N}{n_+}} \sum_{k=1}^{\binom{N}{n'_+}} \langle \beta_{n_+,l} | \sum_{x=1}^{N} \sigma_1(x) | \beta_{n'_+,k} \rangle$$

$$= \frac{1}{\sqrt{\binom{N}{n_+}}} \frac{1}{\sqrt{\binom{N}{n'_+}}} \sum_{l=1}^{\binom{N}{n_+}} \sum_{k=1}^{\binom{N}{n'_+}} \langle \beta_{n_+,l} | \left( \sum_{x \in W^1_{n'_+}} + \sum_{x \in W^2_{n'_+}} \sigma_1(x) \right) | \beta_{n'_+,k} \rangle =$$

$$\frac{1}{\sqrt{\binom{N}{n_+}}} \frac{1}{\sqrt{\binom{N}{n'_+}}} \left( \binom{N}{n_+} (N - n_+) \langle \beta_{n_+,l} | \beta_{n'_+ - 1,k} \rangle + \binom{N}{N - n_+} n_+ \langle \beta_{n_+,l} | \beta_{n'_+ + 1,k} \rangle \right) =$$

$$\sqrt{(N - n_+)(n_+ + 1)} \, \delta_{n_+, n'_+ - 1} + \sqrt{n_+(n_- + 1)} \, \delta_{n_+, n'_+ + 1}. \tag{2.14}$$

We used the fact that the vectors $\beta_{n'_+,l}$ are orthonormal, that

$$\frac{1}{\sqrt{\binom{N}{n_+}}} \frac{1}{\sqrt{\binom{N}{n'_+}}} \binom{N}{n_+} (N - n_+) = \sqrt{(N - n_+)(n_+ + 1)}, \tag{2.15}$$

with $n'_+ - 1 = n_+$, and that

$$\frac{1}{\sqrt{\binom{N}{n_+}}} \frac{1}{\sqrt{\binom{N}{n'_+}}} \binom{N}{N - n_+} n_+ = \sqrt{n_+(N - n_+ + 1)}, \tag{2.16}$$

with $n'_+ + 1 = n_+$. Hence the matrix entries of $h_N^{(2)}$ written with respect to the symmetric basis vectors $|n_+, N - n_+\rangle$, are given by $\sqrt{(N - n_+)(n_+ + 1)}$ on the upper diagonal and by $\sqrt{n_+(N - n_+ + 1)}$ on the lower diagonal (see also [34], §3.1). $\qquad\square$

From now on we will denote the Curie–Weiss Hamiltonian (2.1) represented in the symmetric basis by $J_{N+1}$.

## 2.2 Numerical simulations

In the next section we will argue that for $0 < B < 1$ the above $(N + 1)$-dimensional matrix, denoted by $J_{N+1}$, can be linked to a Schrödinger operator with a symmetric double well on $L^2([0,1])$, for $N$ sufficiently large.[9] Since for a sufficiently high and broad enough

---

[9]Mapping quantum spin systems onto Schrödinger operators is not new, see e.g. [35,36]. Schrödinger operators and quantum spin systems also meet in the large research field of Anderson localization and more generally random Schrödinger operators, see e.g. [37] for a rigorous approach.

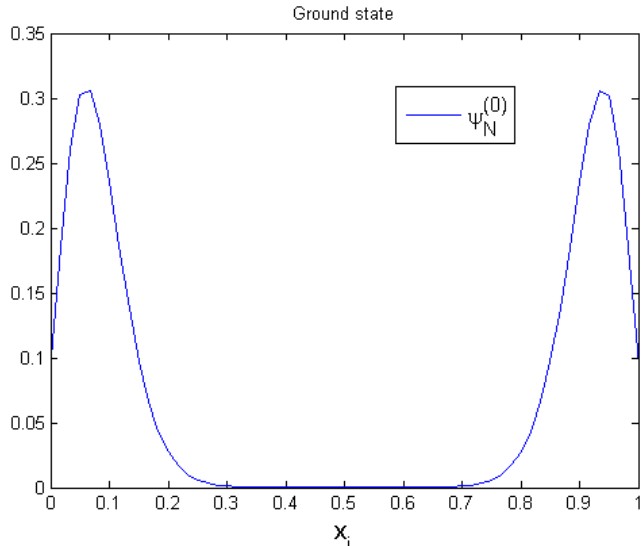

*Figure 1: Ground state eigenfunction of $h_N^{CW}$, computed from the tridiagonal matrix $J_{N+1}$ for $N = 60$, $J = 1$ and $B = 1/2$. The grid points on the horizontal axis are labeled by $x_i = i/N$ for $i = 0, ..., N$.*

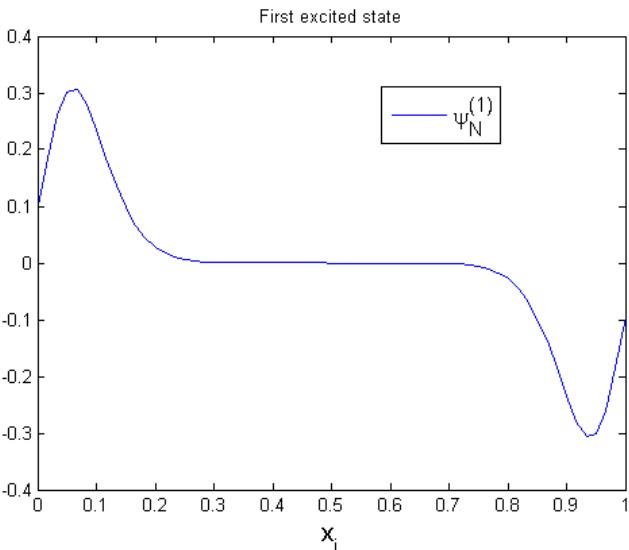

*Figure 2: First excited state of $h_N^{CW}$, computed from the tridiagonal matrix $J_{N+1}$ for $N = 60$, $J = 1$ and $B = 1/2$. The grid points on the horizontal axis are indicated by $x_i = i/N$ for $i = 0, ..., N$.*

potential barrier the ground state of such a Schrödinger operator is approximately given by two Gaussians, each of them located in one of the wells of the potential, we might expect the same result for $J_{N+1}$. In fact, the first two eigenfunctions of this Schrödinger operator are approximately given by

$$\psi^{(0)} \cong \frac{T_a(\varphi_0) + T_{-a}(\varphi_0)}{\sqrt{2}}; \qquad \psi^{(1)} \cong \frac{T_a(\varphi_0) - T_{-a}(\varphi_0)}{\sqrt{2}}, \qquad (2.17)$$

where $T_{\pm a}$ is the translation operator over distance $a$ (i.e., $(T_{\pm a}\varphi_0)(x) = \varphi_0(x \pm a)$), where $\pm a$ denotes the minima of the potential well. The functions $\varphi_n$ are the weighted

Hermite polynomials given by $\varphi_n(x) = e^{-x^2/2} H_n(x)$, with $H_n$ the Hermite polynomials. We diagonalized the operator $J_{N+1}$ and plotted the first two (discrete) eigenfunctions $\psi_N^{(0)}$ and $\psi_N^{(1)}$. For convenience, we scaled the grid to unity. See Figures 1 and 2. From these two plots, it is quite clear that both eigenvectors of $h_N^{CW}$ are approximately given by (2.17). In the following discussion about numerical (in)accuracy, we assume that not only the ground state $\psi_N^{(0)}$ (for which the claim is a theorem) but also the first excited state $\psi_N^{(1)}$ lies in the symmetric subspace $\text{Sym}^N(\mathbb{C}^2)$.[10] For $N \leq 60$ (a number that obviously depends on the machine precision

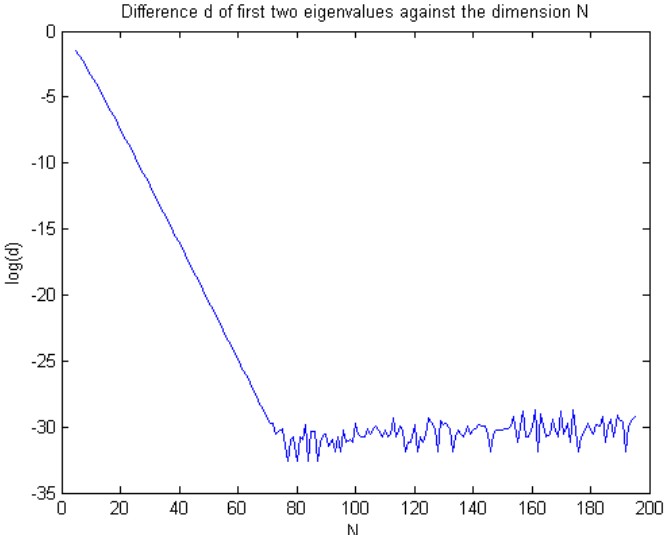

Figure 3: Energy splitting $d = |\epsilon_N^{(0)} - \epsilon_N^{(1)}|$ between the first two eigenvalues $\epsilon_N^{(0)}$ and $\epsilon_N^{(1)}$ of the tridiagonal matrix $J_{N+1}$, for different values of $N$ on a log scale. From about $N = 80$ onwards, the energy splitting is in the order of the maximum achievable accuracy of the eigenvalues, so that the first two eigenvalues become numerically degenerate.

of the computer used to perform the numerical simulations) our simulations accurately reflect the uniqueness of the ground state, proved from general principles. However, for larger values of $N$, clearly visible from about $N \geq 80$ (see Figure 3), up to numerical precision the exact (and unique) ground state is joined by a degenerate state, and the selection of any specific linear combination of those as the numerically obtained "ground state" is implicitly made by numerical noise playing a role similar to some symmetry-breaking perturbation (be it the flea perturbation in section 4 or a more traditional symmetry-breaking field typically used in quantum spin systems); this noise then localizes the ground state purely due to the inaccuracy of the computation.

Consequently, plotting the ground state and the first excited state of $h_N^{CW}$ (at $B = 1/2$ and $J = 1$) for $N \geq 80$ gives a Gaussian curve, located in one of the wells.[11] Specifically, the new

---

[10]This second claim is not essential for our results themselves but is helpful for the following explanation thereof. The point is that the first excited state computed from the tridiagonal matrix $J_{N+1}$ might not the same as the one from the original Hamiltonian $h_N^{CW}$, in which case Figure 2 would be misleading. Fortunately, we have shown numerically that up to $N = 12$ the first excited state of $h_N^{CW}$ represented as a matrix on $\mathbb{C}^{2^N}$ is the same as the one corresponding to the tridiagonal matrix $J_{N+1}$. For $N > 12$ this computation became unfeasible, as the dimension of the relevant subspace grows exponentially with $N$.

[11]We checked this numerically, but omitted the plots. Moreover, we observed that for increasing $N$ these two numerical degenerate states were randomly localized in (one of the) both wells.

(numerically) degenerate ground state eigenvectors are given by the functions

$$\chi_+ = \frac{\psi_N^{(0)} + \psi_N^{(1)}}{\sqrt{2}}; \qquad\qquad \chi_- = \frac{\psi_N^{(0)} - \psi_N^{(1)}}{\sqrt{2}}. \tag{2.18}$$

Using this result and equation (2.17), it follows by a simple calculation that

$$\chi_+ \cong T_a \varphi_0; \qquad\qquad \chi_- \cong T_{-a} \varphi_0, \tag{2.19}$$

where the functions $\varphi_n(x)$ now have to be understood as functions on a discrete grid. Once again, for $N = 60$, the fact that $\psi_N^{(0)}$ (rather than rather than $\chi_\pm$) is the (doubly peaked) ground state is confirmed by the numerical simulations summarized in Figure 1.

## 3 The Curie–Weiss Hamiltonian as a Schrödinger operator

Discretization is the process of approximating the derivatives in (partial) differential equations by linear combinations of function values $f$ at so-called *grid points*. The idea is to discretize the domain, with $N$ of such grid points, collectively called a *grid*. We give an example in one dimension:

$$\Omega = [0, X], \quad f_i \approx f(x_i), \quad (i = 0, .., N), \tag{3.1}$$

with grid points $x_i = i\Delta$ and grid size $\Delta = X/N$. The symbol $\Delta$ is called the *grid spacing*. Note this the grid spacing is chosen to be constant or uniform in this specific example. For the first order derivatives we have

$$\frac{\partial f}{\partial x}(\bar{x}) = \lim_{\Delta x \to 0} \frac{f(\bar{x} + \Delta x) - f(\bar{x})}{\Delta x} = \lim_{\Delta x \to 0} \frac{f(\bar{x}) - f(\bar{x} - \Delta x)}{\Delta x} = \lim_{\Delta x \to 0} \frac{f(\bar{x} + \Delta x) - f(\bar{x} - \Delta x)}{2\Delta x}. \tag{3.2}$$

These derivatives are approximated with *finite differences*. There are basically three types of such approximations:

$$\left(\frac{\partial f}{\partial x}\right)_i \approx \frac{f_{i+1} - f_i}{\Delta x} \quad \text{(forward difference)}$$

$$\left(\frac{\partial f}{\partial x}\right)_i \approx \frac{f_i - f_{i-1}}{\Delta x} \quad \text{(backward difference)}$$

$$\left(\frac{\partial f}{\partial x}\right)_i \approx \frac{f_{i+1} - f_{i-1}}{2\Delta x} \quad \text{(central difference)}. \tag{3.3}$$

Since it is more accurate in our case, will focus on the central difference approximation method and apply this to the second order differential operator $d^2/dx^2$.

### 3.1 Locally uniform discretization

In the example above, the grid spacing was chosen to be uniform. Now reconsider this example on the domain $\Omega = [0, 1]$ with uniform grid spacing $\Delta = 1/N$. The second order derivative operator is approximately given by

$$f_i'' = \frac{f_{i-1} - 2f_i + f_{i+1}}{\Delta^2} + O(\Delta^2) \ (i = 1, ..., N-1). \tag{3.4}$$

By throwing away the error term $O(\Delta^2)$ in the above equation, it follows that we can approximate the second derivative operator in matrix form

$$\frac{1}{\Delta^2}\begin{pmatrix} -2 & 1 & & & & \\ 1 & -2 & 1 & & \text{\huge 0} & \\ & \ddots & \ddots & \ddots & & \\ & \text{\huge 0} & 1 & -2 & 1 & \\ & & & 1 & -2 & \end{pmatrix}. \tag{3.5}$$

This matrix is the standard discretization of the second order derivative on a uniform grid consisting of $N$ points of length $\Delta \cdot N$, with uniform grid spacing $\Delta$. In this specific case, we have $\Delta = 1/N$. We denote this matrix also by $\frac{1}{\Delta^2}[\cdots 1 -2\ 1\cdots]_N$.

Suppose now that we are given a symmetric tridiagonal matrix $A$ of dimension $N$ with constant off- and diagonal elements:

$$A = \begin{pmatrix} b & a & & & & \\ a & b & a & & \text{\huge 0} & \\ & \ddots & \ddots & \ddots & & \\ & \text{\huge 0} & a & b & a & \\ & & & a & b & \end{pmatrix}. \tag{3.6}$$

We are going to extract a kinetic and a potential energy from this matrix. We write

$$A = a[\cdots 1\ \frac{b}{a}\ 1\cdots]_N = a[\cdots 1 -2\ 1\cdots]_N + \text{diag}(b+2a), \tag{3.7}$$

where the latter matrix is a diagonal matrix with the element $b+2a$ on the diagonal. It follows that

$$A = T + V, \tag{3.8}$$

for $T = a[\cdots 1 -2\ 1\cdots]_N$, and $V = \text{diag}(b+2a)$. In view of the above, the matrix $T$ corresponds to a second order differential operator. This matrix plays the role of (3.5), but with uniform grid spacing $1/\sqrt{a}$ on the grid of length $N/\sqrt{a}$. Since the matrix $V$ is diagonal, it can be seen as a multiplication operator. Therefore, given such a symmetric tridiagonal matix $A$, we can derive an operator that is the sum of a discretization of a second order differential operator and a multiplication operator. The latter operator is identified with the potential energy of the system. Hence we can identify $A$ with a discretization of a Schrödinger operator.[12]

The next step is to understand what happens in the case where we are given a symmetric tridiagonal matrix with non-constant off- and on-diagonal elements. This is important as we will see, since the Curie–Weiss Hamiltonian, written with respect to the canonical symmetric basis for the subspace $\text{Sym}^N(\mathbb{C}^2)$ of $\mathbb{C}^{2^N} \simeq \bigotimes_{n=1}^N \mathbb{C}^2$, is precisely an example of such a matrix. The question we ask ourselves is if we can link such a matrix to a discretization of a Schrödinger operator as well (see Appendix B for background).

Writing $T = J_{N+1}$, consider the ratios

$$\rho_j = \frac{h_{j-1}}{h_j} = \frac{T_{j+1}}{T_{j-1}} \quad (j = 1,...,N), \tag{3.9}$$

---

[12]Strictly speaking we have to put a minus sign in front of $T$, as the kinetic energy is defined as $-\frac{d^2}{dx^2}$.

with non-uniform grid spacing $h_j$ and $h_{j-1}$. We divide the original tridiagonal matrix $J_{N+1}$ by $N$ for scaling. Thus, we consider $J_{N+1}/N$. If we then compute the distances $h_j$, we see that they are almost all of $O(1)$, except at the boundaries. We will see later that the corresponding Schrödinger operator analog of the matrix $J_{N+1}/N$ will be an operator on a domain of length $L = \frac{1}{N}\sum_{j=1}^{N} h_j$.

First, we compute the ratios $\rho_j$:

$$\rho_j = \frac{T_{j+1}}{T_{j-1}} = \frac{\sqrt{(N-j)(j+1)}}{\sqrt{(N-j+1)j}} = \sqrt{\frac{N-j}{N-j+1}}\sqrt{\frac{j+1}{j}} = \sqrt{\frac{1}{1+\frac{1}{N-j}}}\sqrt{1+\frac{1}{j}}. \tag{3.10}$$

Use the following approximations

$$\sqrt{1+\frac{1}{j}} \approx 1 + O\!\left(\frac{1}{2j}\right) = 1 + O(1/j) \quad \text{and} \tag{3.11}$$

$$\sqrt{\frac{1}{1+1/(N-j)}} \approx 1 - O\!\left(\frac{1}{2(N-j)}\right) = 1 + O\!\left(\frac{1}{N-j}\right), \tag{3.12}$$

and observe that for $j \gg 1$ and $N - j \gg 1$, we have

$$\sqrt{1+\frac{1}{j}} \approx O(1) \quad \text{and} \tag{3.13}$$

$$\sqrt{\frac{1}{1+1/(N-j)}} \approx O(1). \tag{3.14}$$

Moreover, we see that the ratio satisfies

$$\rho_j \approx 1 + O(1/j) + O\!\left(\frac{1}{N-j}\right), \tag{3.15}$$

using the fact that that the big-O notation respects the product, that $O(\frac{1}{j}\frac{1}{N-j}) \leq O(1/j)$, and also $O(\frac{1}{j}\frac{1}{N-j}) \leq O(\frac{1}{N-j})$.

In the next subsection, we will see from numerical simulations that to a good approximation the ground state eigenfunction is a double peaked Gaussian with maxima centered in the minima of some double well potential that we are going to determine. This potential occurs in a discrete Schrödinger operator analog of the matrix $J_{N+1}/N$ for $N$ large, i.e., in the semiclassical limit. Furthermore, we showed by numerical simulations (Figure 4 below) that the width $\sigma$ of each Gaussian-shaped[13] ground state of $J_{N+1}$ located at one of minima of the potential is of order $\sqrt{N}$, and hence that each peak rapidly decays to zero, so that the ground state eigenfunction is approximately zero at both boundaries. In particular, the size of the domain where the peak is non-zero contains $O(\sqrt{N})$ grid points, as we clearly observe from the figure. This is an approximation, since we neglect the (relatively small) function values of the Gaussian that are more than $O(\sqrt{N})$ away from the central maximum. However, this approximation is highly accurate, as the Gaussian decays to zero exponentially. This observation is extremely important, as we will now see.

Let us first focus on the left-located Gaussian. For a point $x_j = j/N$, clearly $j \in O(N)$. Therefore, for $N$ large enough,

$$\rho_j = 1 + O(1/N), \tag{3.16}$$

---

[13]We mean that if we plot the discrete points and draw a line through these points, then the corresponding graph has the shape of a Gaussian.

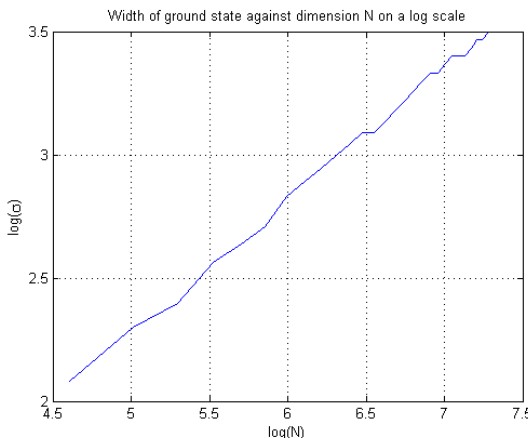

*Figure 4: Width at half height of the ground state eigenvector of $J_{N+1}$ ($B = 1/2$ and $J = 1$) against N, for $N = 100 : 50 : 1500$ on a log scale. The slope of the line is about 0.5, which means that the width $\sigma$ goes like $\sqrt{N}$.*

since for these values of $j < N - j$ we have $O(\frac{1}{N-j}) \leq O(1/j)$. For the right-located peak, we have $N - j < j$, so that in this case $O(1/j) \leq O(\frac{1}{N-j})$, and we find

$$\rho_j = 1 + O\left(\frac{1}{N-j}\right). \tag{3.17}$$

We will now show that, in the present context where we work on a domain of order $L$ (i.e. $[0, L]$), we indeed have uniform discretization on a subinterval of this domain corresponding to a matrix segment of $O(\sqrt{N})$ entries. We start with the peak on the left. Since the error per step that we make equals $\rho_j$, it follows that the error on a matrix segment of length of order $\sigma$ equals $\rho_j^\sigma \approx (1 + \frac{1}{N})^\sigma$ for $j < N - j$ and $N$ large. Denoting the off-diagonal element corresponding to the minimum $x_{j_0}$ of the potential well by $T_{j_0}$, for the off-diagonal elements within a range of order $\sigma$, we derive the next estimate:

$$|T_{j_0} - T_{j_0+\sigma}| \approx |T_{j_0} - O\left((1 + \frac{1}{N})^\sigma\right)T_{j_0}| = T_{j_0}|1 - O\left((1 + \frac{1}{N})^\sigma\right)| \leq C\frac{\sigma}{N}, \tag{3.18}$$

where we used the inequality $(1 + 1/N)^\sigma \leq 1 + C\frac{\sigma}{N}$ as well as the fact that $T_{j_0}$ is of order 1. Here, $C > 1$ is a constant independent of $N$. Since the left peak of the Gaussian eigenfunction is approximately non-zero corresponding to a matrix segment of length of order $\sqrt{N}$, we apply the above estimate to $\sigma \approx \sqrt{N}$. We see immediately that $|T_{j_0} - T_{j_0+\sigma}|$ goes to zero. Therefore, on matrix segment of length of order $\sqrt{N}$ centered around the left minimum $x_{j_0}$ of the potential, the off-diagonal elements coincide in the limit $N \to \infty$. This means that the grid spacing becomes constant and hence that we have locally uniform discretization of the domain. By symmetry, the same is true for the peak located on the right of the well. We conclude that for large $N$ the tridiagonal matrix locally behaves like a kinetic energy, and therefore like a discretized Schrödinger operator. All this will be explained in more detail in the next subsection.

## 3.2 Link with a Schrödinger operator

Our aim is to show that for $N$ large enough, the matrix $J_{N+1}/N$ obtained from the Curie–Weiss Hamiltonian by reduction (see §2) locally approximates a discretization matrix representing a Schrödinger operator describing a particle moving in a symmetric double well. This means

that there exists a sub-block of $J_{N+1}/N$ that has a form approximately given by the sum of $-\frac{1}{h^2}[\cdots 1 -2 \ 1 \cdots]_{N+1}$ (for a certain $h$) and a diagonal matrix playing the role of a potential. We started with the symmetric tridiagonal matrix $J_{N+1}/N$ with non-constant entries. In order to link this matrix to a second derivative and a multiplication operator, we needed to apply the non-uniform discretization procedure.

At first sight the off-diagonal matrix entries of $J_{N+1}/N$ cannot immediately be identified with a second order derivative operator, but we have seen that in the limit $N \rightarrow \infty$ we do have uniform discretization on some interval of a length scale corresponding to a segment of $O(\sqrt{N})$ matrix entries. Consequently, for sufficiently large $N$ this discretization becomes approximately uniform on this length scale. From this, we are now going to extract a matrix of the form (3.6) corresponding to a Schrödinger operator on $L^2([0,1])$. We first consider the $(N \times N)$-matrix $H_N$, defined by

$$H_N = T_N + V_N, \tag{3.19}$$

where at $x = n_+/N$ we define

$$T_N(x) = -\frac{1}{d(x)^2}[\cdots 1 -2 \ 1 \cdots]_N, \tag{3.20}$$

keeping in mind that $d(x)$ varies per entry, and $d(x)$ is defined by

$$d(x) = \frac{1}{\sqrt{B}} \frac{1}{((1-x)x)^{1/4}}. \tag{3.21}$$

$V_N$ is a diagonal matrix (and hence a multiplication operator, as in the continuum) given by

$$V_N(x) = -\frac{1}{2}(2x-1)^2 - B\left(\sqrt{(1-x)(x+\frac{1}{N})} + \sqrt{(1-x+\frac{1}{N})x}\right). \tag{3.22}$$

Note that $d(x)$ corresponds to the non-uniform grid spacing. We rewrite:

$$H_N = -\frac{1}{N^2} \frac{1}{(d(x)/N)^2}[\cdots 1 -2 \ 1 \cdots]_N + V_N \tag{3.23}$$

and hence the total length of the interval is given by $\frac{1}{N}\sum_{n_+} d(n_+/N)$. So if $N \rightarrow \infty$, it follows that

$$L := \int_0^1 d(t)dt = \int_0^1 \frac{1}{\sqrt{B}} \frac{1}{((1-t)t)^{1/4}}dt = \frac{2\Gamma[3/4]^2}{\sqrt{B}\sqrt{\pi}}. \tag{3.24}$$

Moreover, the interval coordinate is then given by

$$D(x) = \int_0^x d(t)dt \in [0,L], \quad \text{for } x \in [0,1]. \tag{3.25}$$

It follows that on each matrix segment of $\sqrt{N}$ entries (for $N$ large), the matrix $H_N$ is an approximation of the following Schrödinger operator on $L^2([0,L])$ with the familiar substitution $\hbar = \frac{1}{N}$:

$$h_1 = -\frac{1}{N^2}\Delta + V_N(x), \tag{3.26}$$

where $x$ is chosen appropriately and the segment describes an interval of length $\frac{d(x)\sqrt{N}}{N} = \frac{d(x)}{\sqrt{N}}$, at location

$$D(x) = \int_0^x \frac{1}{\sqrt{B}} \frac{1}{((1-t)t)^{1/4}}dt, \quad x \in [0,1]; \tag{3.27}$$

in the total interval of length $L$.[14] We saw in the previous section that on length scales of order $d(x)/\sqrt{N}$, the function $d(x)$ is approximately constant so that $T_N$ can be seen as a locally uniform discretization of the second order derivative $\Delta$ on the sub interval approximately given by $[D(x) - d(x)/\sqrt{N}, D(x) + d(x)/\sqrt{N}]$. Therefore, on these length scales the operator $\Delta$ indeed corresponds to a uniform part of $\frac{1}{(d(x)/N)^2}[\cdots 1 -2\ 1 \cdots]_N$.

We see in this section that to a very good approximation the spectral properties of both (a priori different) matrices $J_{N+1}/N$ and $\tilde{H}_{N+1}$, defined in (3.34) below, coincide, improving with increasing $N$. Rescaling the interval $[0, L]$ to unity yields a Schrödinger operator $h$, but now defined on $L^2([0,1])$. Hence $h$ is given by

$$h = -\frac{1}{L^2 N^2}\Delta_y + \tilde{V}_N(y) \ \ (y \in [0,1]), \tag{3.28}$$

where, via the potential $V_N$ defined in (3.22), the potential $\tilde{V}_N$ is defined by

$$\tilde{V}_N(y) = V_N(D^{-1}(yL)) \ \ y \in [0,1]. \tag{3.29}$$

Let us now consider the scaled tridiagonal matrix $J_{N+1}/N$. By definition it follows that the elements on the diagonal, the lower diagonal, and the upper diagonal are given by

$$-\frac{1}{2}(2\frac{n_+}{N} - 1)^2, \qquad -B\sqrt{(1 - \frac{n_+}{N} + \frac{1}{N})\frac{n_+}{N}}, \qquad -B\sqrt{(1 - \frac{n_+}{N})(\frac{n_+}{N} + \frac{1}{N})}, \tag{3.30}$$

respectively. The idea is to split the matrix $J_{N+1}/N$ into two parts, one corresponding to the kinetic energy and the other to the potential energy. However, since the off-diagonal elements of $J_{N+1}/N$ are non-constant and not an even function around $x = 1/2$, we cannot isolate these elements and decompose $J_{N+1}/N$ into two parts. Therefore, we approximate the off-diagonal elements by the function (3.21), which is even in $1/2$. This approximation makes perfect sense in the semi-classical limit, and from this it is clear that $J_{N+1}/N \approx H_{N+1}$, where $H_N$ is defined by (3.19). In the limit $N \to \infty$ we pass to the continuum, so that the discrete points $n_+/N$ are understood as real numbers in $[0,1]$. Therefore, equations (3.21) and (3.25) make sense on $[0,1]$. The operator (3.26) is a Schrödinger operator defined on the space $[0, L]$, whereas the operator (3.28) is obtained by rescaling to the unit interval. Therefore, scaling $H_N$ to unity yields the matrix

$$-\frac{1}{L^2 N^2}\frac{1}{(\frac{d(x)}{LN})^2}[\cdots 1 -2\ 1 \cdots]_N + V_N(D^{-1}(yL)), \tag{3.31}$$

where each segment with order $\sqrt{N}$ matrix entries now describes an interval of length $d(x)/(L\sqrt{N})$ at location $y = D(x)/L$ in $[0,1]$, approximating the Schrödinger operator

$$-\frac{1}{L^2 N^2}\Delta + V_N(x) \ \ \text{with} \ \ x = D^{-1}(yL), \tag{3.32}$$

which indeed coincides with (3.28). The matrix (3.31) corresponds to a non-uniform discretization of $h$, in such a way that it is locally uniform, namely on length scales of order $\sqrt{N}$. Finally, discretizing $\Delta_y$ on the unit interval with uniform grid spacing $1/N$ yields the following discretization matrix:

$$K_N = -\frac{1}{L^2 N^2}\frac{1}{(\frac{1}{N})^2}[\cdots 1 -2\ 1 \cdots]_N = -\frac{1}{L^2}[\cdots 1 -2\ 1 \cdots]_N. \tag{3.33}$$

---

[14]Namely, if $z \in [0, L]$, then $x = D^{-1}(z)$. Hence the coordinate $z$ corresponds to a location in the interval $[0, L]$, whilst $x$ plays the role of the 'argument' of a matrix entry of $V$, e.g. if $x = n_+/N$, then $V(x) = V(n_+/N) \equiv V_{n_+}$.

<sup>345</sup> Using this discretization of $-\frac{1}{L^2 N^2}\Delta_y$, we show that for $N \to \infty$ the spectrum of the matrix
<sup>346</sup> $J_{N+1}/N$ approximates the spectrum of the matrix

$$\tilde{H}_N = K_N + \tilde{V}_N. \tag{3.34}$$

<sup>347</sup> Using the fact that $\tilde{H}_N$ is a discretization of (3.28), this indeed establishes a link between the
<sup>348</sup> compressed Curie–Weiss Hamiltonian and a Schrödinger operator.[15]

<sup>349</sup> *Remark.* Consider the Schrödinger operator with a symmetric double well potential, given by
<sup>350</sup> (1.1). Recall from §2.2 that for a sufficiently high and broad potential well, the ground state
<sup>351</sup> of such a Schrödinger operator is approximately given by two Gaussians, each of them located
<sup>352</sup> in one of the wells of the potential (see [38]). This fact will be useful for the next round of
<sup>353</sup> observations.

<sup>354</sup> We will now show that the Gaussian-shaped ground state of $J_{N+1}/N$, indeed localizes in both
<sup>355</sup> minima of the potential well $\tilde{V}_N$. To this end, we have made a plot of the scaled potential
<sup>356</sup> $\tilde{V}_N$ from equation (3.29) on a domain of length 1, for $B = 1/2$ and $J = 1$. See Figure
<sup>357</sup> 5. We immediately recognize the shape of a symmetric double well potential. The points
<sup>358</sup> in its domain are given by $y_j = j/N$ for $j = 0,...,N$, and the argument of the potential
<sup>359</sup> is given by $D^{-1}(y_j L)$, as explained before. Then we diagonalized the matrix $J_{N+1}/N$ and
<sup>360</sup> computed the ground state eigenvector. We plot this together with the potential in Figure
<sup>361</sup> 5. One should mention that only one Gaussian peak is visible, not two. As we have seen in
<sup>362</sup> §2.2, this must be due to the finite precision of the computer i.e., the first two eigenvalues are
<sup>363</sup> already numerically degenerate. Thus the computer picks a linear combination of the first two
<sup>364</sup> eigenvectors as ground state (viz. (2.18)), even though we know from the Perron-Frobenius
<sup>365</sup> Theorem (Appendix A) that the ground state is always unique for any finite $N$.[16] We also
<sup>366</sup> observe that the maxima of the Gaussian ground state peaks are precisely centered in the
<sup>367</sup> minima of these two wells (as should be the case). It is clear from this figure that the ground
<sup>368</sup> state is localized in (one of) the minima of the double well.

<sup>369</sup> One might suggest that there would be some critical value of $N$ for which the eigenvalues are
<sup>370</sup> not yet degenerate for the computer. We have seen in Figure 3 that this value of $N$ (depending
<sup>371</sup> on our machine) is about $N = 80$. Figure 6 is a similar plot for the ground state for $N = 60$, on
<sup>372</sup> a par with Figure 1 in §2.2. We recognize the well-known doubly peaked Gaussian shape, but
<sup>373</sup> now it is localized in both minima of the potential well. This is displayed in Figure 6. These
<sup>374</sup> figures show that there is a convincing relation between the matrix $J_{N+1}/N$ and a Schrödinger
<sup>375</sup> operator describing a particle in a double well. The double well shaped potential is a result of
<sup>376</sup> the choice $B = 1/2$. The value of the magnetic field needs to be within $[0, 1)$ in order to get
<sup>377</sup> spontaneous symmetry breaking of the ground state in the classical limit $N \to \infty$. For $B \geq 1$
<sup>378</sup> the Curie–Weiss model will not display SSB, not even in the classical limit. For these values of
<sup>379</sup> $B$, the well will be a single potential, as depicted in Figure 7 for $B = 2$.

<sup>380</sup> In view of the corresponding Schrödinger operator, the ground state in the classical limit will
<sup>381</sup> not break the symmetry for a single potential well, and is therefore also compatible with the
<sup>382</sup> Curie–Weiss model for $B \geq 1$. We now return to the regime $0 \leq B < 1$. One can compute
<sup>383</sup> the spectral properties of the matrix $J_{N+1}/N$ and compare them with those of the matrix $\tilde{H}_N$
<sup>384</sup> corresponding to the sum of the uniform discretization $K_N$ of the second order derivative (viz.
<sup>385</sup> (3.33)) and $\tilde{V}_N$ (viz. (3.29)). We will see that to a very good approximation the spectral
<sup>386</sup> properties of both matrices coincide and get better with increasing $N$. We have programmed
<sup>387</sup> the matrix $\tilde{H}_N$ in MATLAB. The matrix has been diagonalized. The spectral properties have

---

[15]Note that the matrix $\tilde{H}_N$ corresponding to (3.28) is by definition a discretization of the Schrödinger operator on the whole of $[0, 1]$, and not only for subintervals of length $d(x)/L\sqrt{N}$.

[16]Due to this degeneracy, the computer picks or the symmetric combination $\chi_+$, or the anti-symmetric combination $\chi_-$, which depends on the algorithm. We changed $N$ and observed that the location of the peak changed as well. This suggests that random superpositions of the two degenerate states are formed.

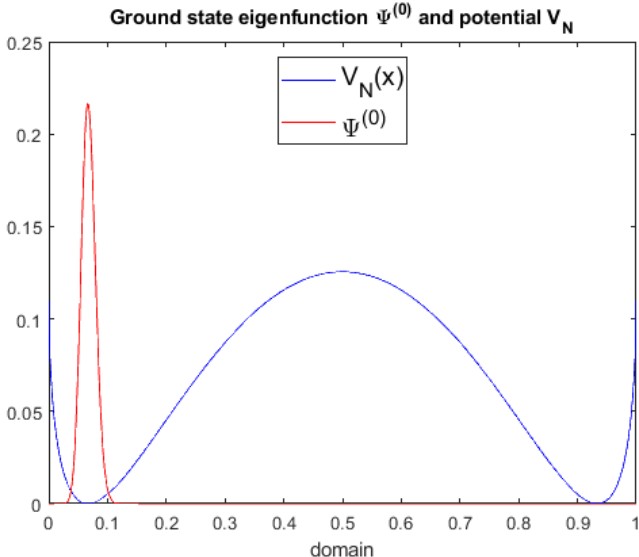

Figure 5: *Scaled potential $V_N := \tilde{V}_N$ and the ground state eigenfunction corresponding to $J_{N+1}/N$ for $N = 1000$ on the unit interval $[0,1]$, plotted on uniform grid points. The potential is shifted so that its minimum is zero, and is plotted on the grid points $x$ corresponding to the solution of $y = D(x)/L$, as explained in the main text.*

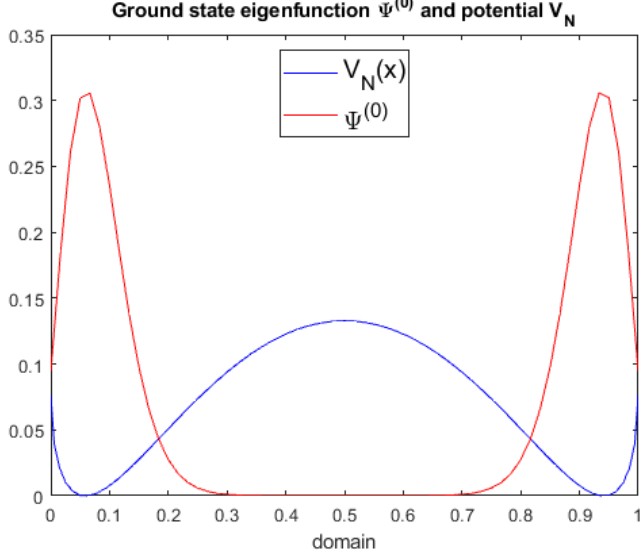

Figure 6: *Scaled and shifted potential $V_N := \tilde{V}_N$ for $B = 1/2$ and $J = 1$, plotted on grid points that are solution of the equation $y = D(x)/L$, and the ground state eigenfunction corresponding to $J_{N+1}/N$ for $N = 60$, plotted on uniform grid points.*

been compared to those of $J_{N+1}/N$ (Table 1). We computed the first ten eigenvalues of the matrix $J_{N+1}/N$, denoted by $\epsilon_n$, and those of $\tilde{H}_N$, denoted by $\lambda_n$. In the left column the eigenvalues $\epsilon_n$ are displayed. In the right column the absolute difference $|\lambda_n - \epsilon_n|$ is displayed. The number $N = 1000$ is fixed (the numerical eigenvalues in the table are dimensionless, since we put $J = 1$, cf. footnote 7).

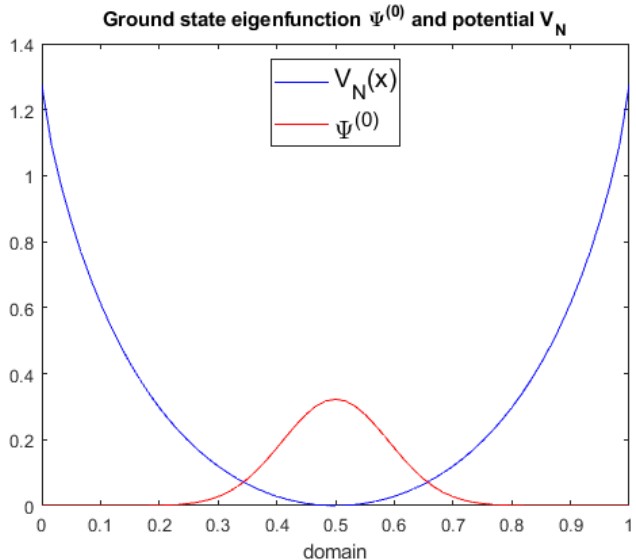

Figure 7: Scaled and shifted potential $V_N := \tilde{V}_N$ for $B = 2$ and $J = 1$, and the ground state eigenfunction corresponding to $J_{N+1}/N$ for $N = 60$. The single well is clearly visible. The ground state is plotted on the uniform grid corresponding to $[0,1]$, and also now the ground state eigenvector is normalized to 1.

Table 1. Eigenvalues and absolute differences ($J = 1, B = 1/2$)

| n | $\epsilon_n$ | $|\lambda_n - \epsilon_n|$ |
|---|---|---|
| 0 | -0.6251 | $7.6038 \times 10^{-7}$ |
| 1 | -0.6251 | $7.6038 \times 10^{-7}$ |
| 2 | -0.6234 | $2.2446 \times 10^{-6}$ |
| 3 | -0.6234 | $2.2446 \times 10^{-6}$ |
| 4 | -0.6217 | $4.8378 \times 10^{-6}$ |
| 5 | -0.6217 | $4.8378 \times 10^{-6}$ |
| 6 | -0.6200 | $7.0222 \times 10^{-6}$ |
| 7 | -0.6200 | $7.0222 \times 10^{-6}$ |
| 8 | -0.6183 | $8.8010 \times 10^{-6}$ |
| 9 | -0.6183 | $8.8010 \times 10^{-6}$ |

We see that the first ten eigenvalues for both matrices are the same up to at least six decimals. It is also clear that these eigenvalues are doubly degenerate, at least up to six decimals. Moreover, we plotted all the eigenvalues $\epsilon_n$ and $\lambda_n$ corresponding to the bound states, i.e., the energy levels within the well. This is depicted in Figure 8 below.

It follows that the energies of both systems are approximately the same. Moreover, we compared a plot of the ground state eigenvector of $J_{N+1}/N$ with the one corresponding to $\tilde{H}_N$, Completely analogously to $J_{N+1}/N$, we observed also now that the ground state of $\tilde{H}_N$ is located in the minima of the potential well, only concentrates on length scales of order $\sqrt{N}$, and exponentially decays to zero. This is in agreement with the theory of Schrödinger operators, since $\tilde{H}_N$ represents a discretization of a Schrödinger operator as we have seen in the beginning of this section. Table 1, Figure 8 and this observation show that we have strong numerical evidence that the original tridiagonal matrix is related to $\tilde{H}_N$, which was *a priori* not clear since the off-diagonal elements of $J_{N+1}/N$ are non-constant.

*Remark.* It is well known that the ground state of the operator $h$ for finite $N$ looks approximately like a doubly peaked Gaussian, where each peak is centered in one of the minima of the

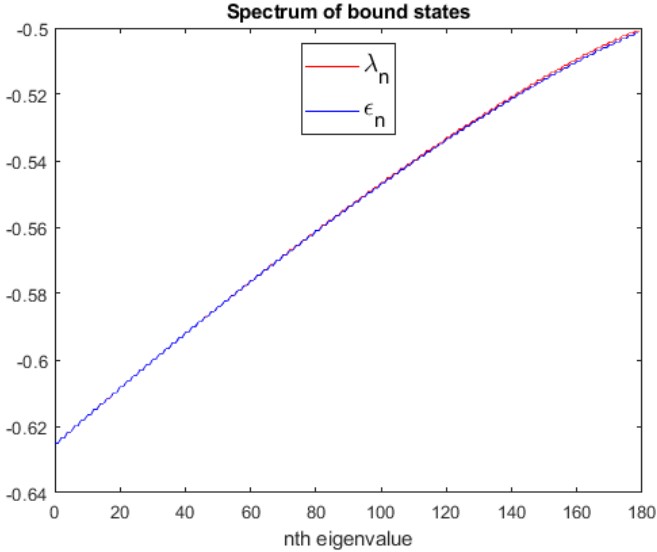

Figure 8: *Spectrum of the bound states in the unscaled potential $\tilde{V}_N$ for $B = 1/2$, $J = 1$, and $N = 1000$; $\epsilon_n$ corresponds to the eigenvalues of $J_{N+1}/N$ and $\lambda_n$ to those of $\tilde{H}_N$.*

potential. For infinite $N$, these peaks will behave like delta distributions. Moreover, numerical simulations (Figure 4) show that the eigenfunctions of $\tilde{H}_N$ live approximately on a grid of order $\sqrt{N}$ points on the interval $[0, 1]$. Using the above discretization, we then have about $\sqrt{N}$ steps of $1/N$ each, so that in particular the ground state Gaussian has a width of $1/\sqrt{N}$. On the one hand, it is clear that this width will go to zero as $N \to \infty$. On the other hand, also the unit interval depends on $N$, as the latter has to be discretized with $N + 1$ points. Therefore, the total number of grid points in the ground state peak living on a subset of order $\sqrt{N}$ is

$$\frac{1/\sqrt{N}}{1/N} = \sqrt{N}. \tag{3.35}$$

In fact, due to the discretization of the grid we have a better approximation of the Gaussian ground state when $N$ increases.

We have computed the minimum of the potential, and subtracted this minimum from the lowest eigenvalues. Then, we have set the potential minimum to zero. These shifted eigenvalues then live in a positive potential well. For $J_{N+1}/N$ with $N = 1000$, we now consider its eigenvalues $\epsilon_n$. We have already seen above that the lowest eigenvalues of $J_{N+1}/N$ become doubly degenerate. Therefore, we identify these approximately doubly degenerate eigenstates with one single state that we denote by $n$. It follows that each $n$ corresponds to two (approximately) degenerate eigenvalues, e.g., $n = 0$ corresponds to the ground state as well as the first excited state of $J_{N+1}/N$, $n = 1$ corresponds to the second and the third excited state, and so on. This is displayed in table 2 below.

Table 2. Shifted eigenvalues for odd values of n ($J = 1, B = 1/2$)

| n | $\epsilon_n$ |
|---|---|
| 0 | 0.000863 |
| 1 | 0.002591 |
| 2 | 0.004310 |
| 3 | 0.006013 |
| 4 | 0.007710 |

Using this table, we deduce that when $N$ is large enough the energy splitting is approximately given by $\sqrt{3}/N$. The ground state (shifted) eigenvalue (which is approximately doubly

degenerate) is then given by $\frac{1/2\sqrt{3}}{N}$, the first excited state (also approximately doubly degenerate) is $\frac{3/2\sqrt{3}}{N}$, the second excited state is $\frac{5/2\sqrt{3}}{N}$ etc. Therefore, there is excellent numerical evidence that for sufficiently large $N$ and $J = 1, B = 1/2$ the (approximately) doubly degenerate shifted spectrum of $J_{N+1}/N$ is given by

$$\frac{(n+1/2)\sqrt{3}}{N}, \quad (n = 0, 1, 2, ...). \tag{3.36}$$

This firstly shows that for large $N$ the wells approximately decouple since tunneling is suppressed in the limit, and secondly that each well of the double well potential is locally quadratic and therefore approximately has the spectrum of a harmonic oscillator (the latter approximation increasingly breaks down at higher excitation energies, however). Finally, both tables have been computed for fixed $N$, but also different values of $N$ need to be considered. Table 3 shows the ground state eigenvalue $\epsilon_0^N$ of the matrix $J_{N+1}/N$.

Table 3. $N\epsilon_0^N$ for increasing $N$ ($J = 1, B = 1/2$)

| N | $N\epsilon_0^N$ |
|---|---|
| 100 | 0.8473 |
| 1000 | 0.8633 |
| 2500 | 0.8653 |
| 5000 | 0.8655 |

Thus $\epsilon_0^N$ will approximate $\frac{1/2\sqrt{3}}{N}$ when $N$ increases, which confirms (3.36).

## 4 Symmetry breaking in the Curie–Weiss model

In this section we introduce a perturbation in the quantum Curie–Weiss model $h_N^{CW}$ such that the symmetric and hence delocalized ground state as displayed in Figure 1 breaks the $\mathbb{Z}_2$ symmetry and hence localizes already for finite (but large) $N$. To find the appropriate perturbation of the Curie–Weiss Hamiltonian, let us first continue the discussion in the Introduction by reviewing the flea perturbation of the symmetric double well potential, following Ref. [1, 2, 26].

### 4.1 Review of the "flea" perturbation on the double well potential

Generalizing (1.2), consider a one-dimensional Schrödinger operator

$$h_\hbar = -\hbar^2 \frac{d^2}{dx^2} + V(x), \tag{4.1}$$

where the potential $V$ is $C^\infty$, non-negative, strictly positive at $\infty$, zero at two points $m_1$ and $m_2 > m_1$, and $\mathbb{Z}_2$-symmetric. Then consider the *Agmon metric* $d_V$ on $\mathbb{R}$, defined by

$$d_V(x, y) = \int_x^y \sqrt{V(s)} ds. \tag{4.2}$$

A *flea perturbation* $\delta V$ is $C^\infty$, non-negative, bounded, and such that $\delta V(x) = 0$ in neighbourhoods of the minima $m_1$ and $m_2$ of $V$. Nonetheless, its support should be close to one of these minima, in the following sense. We use the following notation:

- $d_0 = d_V(m_1, m_2)$ is the Agmon distance between the two minima of $V$;

457   • $d_1 = 2\min\{d_V(m_1, \operatorname{supp}\delta V), d_V(m_2, \operatorname{supp}\delta V)\}$ is twice the Agmon distance between
458      the support of $\delta V$ and the minimum that is closest to to this support;

459   • $d_2 = 2\max\{d_V(m_1, \operatorname{supp}\delta V), d_V(m_2, \operatorname{supp}\delta V)\}$ is twice the Agmon distance between
460      the support of $\delta V$ and the minimum that is furthest away from this support.

461 Finally, and perhaps most crucially, as $\hbar \to 0$ the perturbation should dominate the energy
462 difference $\Delta(E)_\hbar = E_1(\hbar) - E_0(\hbar)$ between the ground state of the unperturbed Hamiltonian
463 (4.1) and the first excited state. But since $\Delta(E)_\hbar \sim \exp(-d_0/\hbar)$, satisfying this condition is a
464 piece of cake. Detailed analysis [2] then shows that:

465   • If $d_0 < d_1 \leq d_2$ there is no localization of the ground state.

466   • If $d_1 < d_0 \leq d_2$ the ground state localizes near the minimum $m_i$ furthest from the support
467      of $\delta V$ (if $\delta V$ were negative, it would localize closest to the support of $\delta V$);

468   • If $d_1 < d_2 < d_0$ the ground state localizes as in the previous case.

469 Though perhaps surprising at first sight, this is actually easy to understand, either from
470 energetic considerations or from a $2 \times 2$ matrix analogy [2,7]. First, the ground state tries
471 to minimize its energy according to the rules:

472   • The cost of localization (if $\delta V = 0$) is $\mathcal{O}(e^{-d_0/\hbar})$.

473   • The cost of turning on $\delta V$ is $\mathcal{O}(e^{-d_1/\hbar})$ when the wave-function is delocalized.

474   • The cost of turning on $\delta V$ is $\mathcal{O}(e^{-d_2/\hbar})$ when the wave-function is localized in the well
475      around $x_0 = m_i$ for which $d_V(m_i, \operatorname{supp}\delta V) = d_2$ $(i = 1, 2)$.

476 For the latter, define a 2-level Hamiltonian

$$h_\hbar^{(2)} = \tfrac{1}{2}\begin{pmatrix} 0 & -\Delta(E)_\hbar \\ -\Delta(E)_\hbar & 0 \end{pmatrix}. \tag{4.3}$$

477 The eigenvalues and eigenvectors of $h_\hbar^{(2)}$, respectively, are given by

$$E_0(\hbar) = -\tfrac{1}{2}\Delta(E)_\hbar \qquad\qquad \varphi^{(0)} = \frac{1}{\sqrt{2}}\begin{pmatrix} 1 \\ 1 \end{pmatrix}; \tag{4.4}$$

$$E_1(\hbar) = -\tfrac{1}{2}\Delta(E)_\hbar \qquad\qquad \varphi^{(0)} = \frac{1}{\sqrt{2}}\begin{pmatrix} 1 \\ -1 \end{pmatrix}. \tag{4.5}$$

478 Hence $E_1(\hbar) - E_0(\hbar) = \Delta(E)_\hbar$, and the (metaphorically) localized states would be

$$\varphi^+ \equiv \tfrac{1}{2}\left(\varphi^{(0)} + \varphi^{(1)}\right) = \begin{pmatrix} 0 \\ 1 \end{pmatrix}, \quad \varphi^- \equiv \tfrac{1}{2}(\varphi^{(0)} - \varphi^{(1)}) = \begin{pmatrix} 1 \\ 0 \end{pmatrix}. \tag{4.6}$$

479 Now take $\delta > 0$ and introduce a "flea" perturbation by changing $h_\hbar^{(2)}$ to $h_\hbar^{(2)} + \delta^{(2)}V$, where

$$\delta^{(2)}V = \begin{pmatrix} 0 & 0 \\ 0 & \delta \end{pmatrix}. \tag{4.7}$$

480 The eigenvalues of $h_\hbar^{(2)} + \delta^{(2)}V$ then shift from $(E_0(\hbar), E_1(\hbar))$ to $(E_-(\hbar), E_+(\hbar))$, where

$$E_\pm(\hbar) = \tfrac{1}{2}\left(\delta \pm \sqrt{\delta^2 + \Delta(E)_\hbar^2}\right), \tag{4.8}$$

with corresponding normalized eigenvectors $(\psi_\hbar^-, \psi_\hbar^+)$ given by

$$\psi_\hbar^- = \frac{1}{\sqrt{2}}\left(\delta^2 + \Delta(E)_\hbar^2 + \delta\sqrt{\delta^2 + \Delta(E)_\hbar^2}\right)^{-1/2}\begin{pmatrix} \Delta(E)_\hbar \\ \delta + \sqrt{\delta^2 + \Delta(E)_\hbar^2} \end{pmatrix}; \qquad (4.9)$$

$$\psi_\hbar^+ = \frac{1}{\sqrt{2}}\left(\delta^2 + \Delta(E)_\hbar^2 - \delta\sqrt{\delta^2 + \Delta(E)_\hbar^2}\right)^{-1/2}\begin{pmatrix} \Delta(E)_\hbar \\ \delta - \sqrt{\delta^2 + \Delta(E)_\hbar^2} \end{pmatrix}. \qquad (4.10)$$

As long as $\lim_{\hbar \to 0} \Delta(E)_\hbar/\delta = 0$, we have

$$\lim_{\hbar \to 0} \psi_\hbar^\pm = \varphi^\pm, \qquad (4.11)$$

so that under the influence of the flea perturbation the ground state localizes as $\hbar \to 0$. There is no need for a separate limit $\delta \to 0$, since it follows from the limit $\hbar \to 0$.

Returning to the real thing, a (mathematically) very natural flea-like perturbation $\delta V$ for the Schrödinger operator $h_\hbar$, and the one we shall mimic for the Curie–Weiss model, is

$$\delta V_{b,c,d}(x) = \begin{cases} d \exp\left[\frac{1}{c^2} - \frac{1}{c^2-(x-b)^2}\right] & \text{if } |x-b| < c \\ 0 & \text{if } |x-b| \geq c \end{cases}, \qquad (4.12)$$

where the parameters $(b,c,d)$ represent the location of its center $b$, its width $2c$ and its height $d$, respectively. Tuning these, the conditions above can be satisfied in many ways: for example, if $b > c > m_2$ the condition $d_1 < d_0 \leq d_2$ for asymmetric localization reads

$$2\int_{m_2}^{b-c}\sqrt{V(s)} < \int_{m_1}^{m_2}\sqrt{V(s)}ds \leq 2\int_{m_1}^{b-c}\sqrt{V(s)}, \qquad (4.13)$$

which can be satisfied by putting $b$ close to $m_2$ (depending on the central height of $V$).

## 4.2 Peturbation of the Curie–Weiss Hamiltonian

The next step in our analysis, then, is to find an analogous perturbation to (4.12) but now for the Curie–Weiss Hamiltonian, using the fact that the Curie–Weiss model is related to a Schrödinger operator (see previous section). To this end, recall the symmetriser $S_N$ defined in (1.10), which is a projection onto the space of all totally symmetric vectors. As we have seen, a basis for the space of totally symmetric vectors is given by the vectors $\{|n_+, n_-\rangle| \; n_+ = 0, ..., N\}$, which spans the subspace $\text{Sym}^N(\mathbb{C}^2)$. In order to define the symmetry-breaking flea perturbation, again we may pick a basis for $\mathcal{H}_N$ and define the perturbation on a basis for $\mathcal{H}_N$. Since the original Hamiltonian was defined on the standard basis $\beta$, we do the same for the perturbation. In the proof of Theorem 2.1, we have seen there is a bijection between the number of orbits and the dimension of $\text{Sym}^N(\mathbb{C}^2)$, namely

$$\mathcal{O}^k \longleftrightarrow |N-k, k\rangle, \qquad (4.14)$$

where $k$ in $|N-k, k\rangle$ labels the number of occurrences of the vector $e_2$ in any of the basis vectors $\beta_i \in \beta$, and likewise $N-k$ in $|N-k, k\rangle$ labels the number of occurrences of the vector $e_1$ in $\beta_i$, so that $N-k$ stands for the number of spins in the up direction whereas the second position $k$ denotes the number of down spins. By definition, $S_N$ maps any basis vector $\beta_k \in \beta$ in a given orbit $\mathcal{O}^k$ to the same vector in $\text{Sym}^N(\mathbb{C}^2)$, which equals

$$\frac{1}{\sqrt{\binom{N}{k}}}\sum_{l=1}^{\binom{N}{k}}\beta_{k_l}. \qquad (4.15)$$

Here the suffix $l$ in $\beta_{k_l}$ labels the basis vector $\beta_k \in \beta$ within the same orbit $\mathcal{O}^k$. So for each orbit $\mathcal{O}^k$, we have $\binom{N}{k}$ vectors $\beta_k$. Hence for each $l = 1, ..., \binom{N}{k}$ the image $S_N(\beta_{k_l})$ under $S_N$ is always the same, namely the coordinate vector written with respect to $\beta$.

The perturbation we are going to define is very similar to the symmetriser $S_N$. Of course, since we have expressed our original Curie–Weiss Hamiltonian with respect to this $|n_+, n_-\rangle$ basis, we need to do the same for the perturbation. Since we have a partition of our $2^N$-dimensional basis $\beta$ into $N + 1$ orbits, we define our perturbation $\Delta V_N$ by

$$\Delta V_N(\beta) = \delta V_{b,c,d}\left(\frac{k}{N}\right) S_N(\beta), \tag{4.16}$$

where $k \in \{0, ..., N\}$ is the unique number such that $\beta \in \mathcal{O}^k$, and $\delta V_{b,c,d}$ is defined by (4.12). We will see that a specific choice of parameters results in localization of the ground state as $N \to \infty$. First, note that when we transform the matrix $[\Delta V_N]_\beta$ in the $\beta$ basis to the corresponding matrix in the $|n_+, n_-\rangle$ basis, it is obvious that it becomes a diagonal matrix with the value $\Delta V_k \equiv \delta V_{b,c,d}(k/N)$ at entry $(k, k)$, since all basis vectors within the same orbit are mapped to the same vector under $\Delta V_N$. If we can show that

$$[S_N, \Delta V_N] = 0, \tag{4.17}$$

and that the ground state eigenfunction of the perturbed Hamiltonian $h_N + \Delta V_N$ is unique and positive,[17] then we may conclude that the ground state lies in the subspace $\mathrm{Sym}^N(\mathbb{C}^2)$. The reason for this is the same as for the unperturbed Curie–Weiss Hamiltonian: these properties push this eigenvector into the subspace $\mathrm{ran}(S_N) = \mathrm{Sym}^N(\mathbb{C}^2)$, so that we may diagonalize this Hamiltonian represented as a matrix that can be written with respect to the symmetric subspace, which will be a tridiagonal matrix of dimension $N + 1$ as well. This makes computations much easier, and allows one to compare the unperturbed system with the perturbed one. Similarly as for the Curie–Weiss model, a sufficient condition for uniqueness and positivity of the ground state of the perturbed matrix, originally written with respect to the standard basis for $\mathcal{H}_N$, is non-negativity and irreducibility, so that we can apply the Perron–Frobenius Theorem, as explained in Appendix A. This depends, of course, on the parameters of the perturbation $\Delta V_N$. We will come back to this later. In order to prove (4.17), it suffices to do so on a basis, for which we take the standard basis $\beta$ of the $N$-fold tensor product. Fix a basis vector $\beta$ in $\mathcal{O}^k$. Then we immediately find

$$\Delta V_N S_N(\beta) = \delta V_{b,c,d}(k) S_N^2(\beta) = S_N \delta V_{b,c,d}(k) S_N(\beta) = S_N \Delta V_N(\beta), \tag{4.18}$$

which proves (4.17). The last step is to show that the Hamiltonian $-(h_N + \Delta V_N)$, written with respect to the standard basis $\beta$ for $\mathcal{H}_N$, is a non-negative and irreducible matrix. Since the off-diagonal elements are completely determined by the unperturbed Hamiltonian and are never zero, the matrix can never be decomposed into two blocks, so that it remains irreducible. Non-negativity is achieved when

$$\frac{J}{2N}(2n_+ - N)^2 - \Delta V_{n_+} \geq 0, \tag{4.19}$$

which is clearly satisfied for positive $d$. Therefore, taking $d > 0$ together with the fact that $h_N + \Delta V_N$ commutes with $S_N$, shows in the same way as for the unperturbed Curie–Weiss model

---

[17]Equivalently, without using positivity of the eigenfunction one would reach the same conclusion by showing that the ground state is unique and $\Delta V_N$ commutes with the entire permutation group on $N$ elements (like the unperturbed Hamiltonian). Given positivity, it is enough to check commutativity merely with the projection $S_N$, because all nontrivial permutations would transform the ground state wavefunction into a function that is no longer strictly positive. We are indebted to Valter Moretti for this comment.

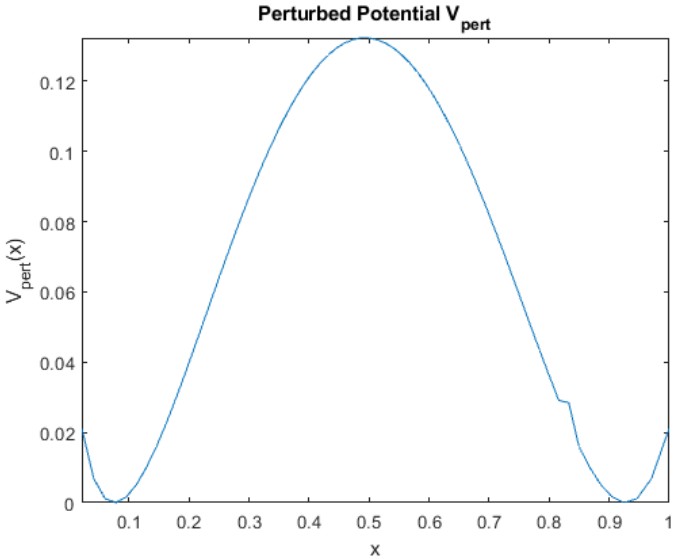

*Figure 9: Perturbed potential computed from the tridiagonal matrix $h_N^{CW} + \Delta V_N$ in the symmetric basis for $N = 65$, $b = (N-9)/N$, $c = 1/45$, $d = 0.4$, $J = 1$ and $B = 1/2$. This potential has a 'flea' on the right side of the well due the perturbation $\Delta V_N$.*

that the ground state of the perturbed Hamiltonian is unique and positive, and therefore lies in $\mathrm{ran}(S_N) = \mathrm{Sym}^N(\mathbb{C}^2)$, where it can be diagonalized.

Recall that in §2.2 the ground state $\psi_N^{(0)}$ of the unperturbed Hamiltonian $h_N^{CW}$ was approximately given by two Gaussians (for $N$ large), each of them located in one of the wells of the potential, and was given by

$$\psi_N^{(0)} \cong \frac{T_a(\varphi_0) + T_{-a}(\varphi_0)}{\sqrt{2}}. \tag{4.20}$$

We now show numerically that our flea perturbation $\Delta V_N$ forces the ground state to localize for large $N$, leaving an analytic proof à la Simon (1985) to the future.

As in section 3, we extract the potential corresponding to the perturbed Hamiltonian $h_N + \Delta V_N$, written with respect to the symmetric basis, scale this Hamiltonian by $1/N$, and translate the potential so that its minima are set to zero. We plot this perturbed potential on the unit interval in Figure 9, where for convenience we scale the domain to the unit interval. Moreover, we plot the ground state of this Hamiltonian and the one corresponding to the unperturbed one in Figure 10, observing localization of the ground state in the left sided well. Numerical simulations show that the eigenvalues of the perturbed Hamiltonian are non-degenerate, so that the ground state is unique, and hence localization is not a result of numerical degeneracy but is genuinely caused by the perturbation. A similar simulation for flea perturbation, but now located on the left site of the barrier, is shown in Figure 11. As expected, we now see a localization of the ground state to the right side of the barrier (Figure 12).

Our conclusion is that due to the flea perturbation, the ground state will localize in one of the wells depending on where the flea is put. As in the continuous Schrödinger operator case, this localization may be understood from energetic considerations: for example, if the perturbation is located on the right, then the relative energy in the left-hand part of the double well is lowered, so that localization will be to the left. This even happens if the perturbation vanishes in the limit $N \to \infty$. We have seen that our (unscaled) tridiagonal matrix $J_{N+1}$ to a

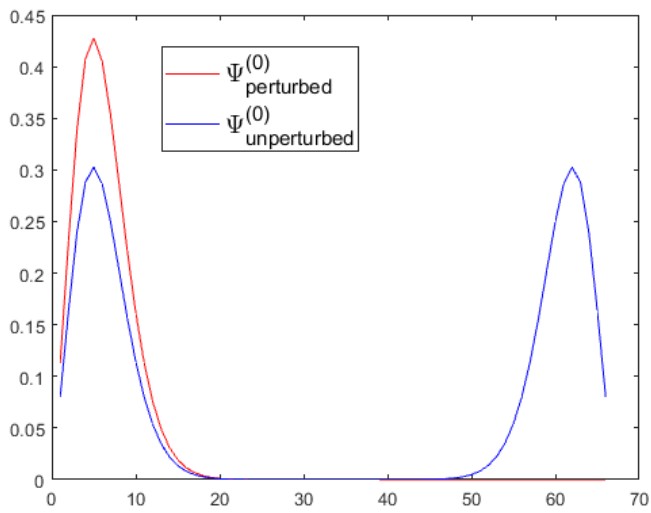

*Figure 10: Corresponding ground state (in red) of the perturbed Hamiltonian $h_N^{CW} + \Delta V_N$ is already localized for $N = 65$, $b = (N-9)/N$, $c = 1/45$, $d = 0.4$, $J = 1$ and $B = 1/2$. Localization takes place on the left side of the well, since the flea raises the potential on the right side.*

very good approximation, is a discretization of the operator $Nh$, where

$$h = -\frac{1}{N^2 L^2}\Delta_y + V, \tag{4.21}$$

where $\Delta_y = d^2/dy^2$ and $V$ the double well potential. It follows that for the perturbed Hamiltonian (where $\Delta V_N = O(1)$ fixed, as in our case)

$$J_N + \Delta V_N \approx N\left(-\frac{1}{(NL)^2}\Delta_y + V\right) + \Delta V_N$$

$$= N\left(-\frac{1}{(NL)^2}\Delta_y + V + \Delta V_N/N\right), \tag{4.22}$$

which implies that the perturbation $\Delta V_N/N$ effectively disappears as $N \to \infty$. According to Jona-Lasinio *et al* (1981) and Graffi *et al* (1984), we can even take

$$\Delta V_N = O(1/N); \tag{4.23}$$

$$\Delta V_N/N = O(1/N^2), \tag{4.24}$$

and also in this case a collapse of the ground state takes place. This is also clear from the $2 \times 2$ matrix argument given in the previous section.

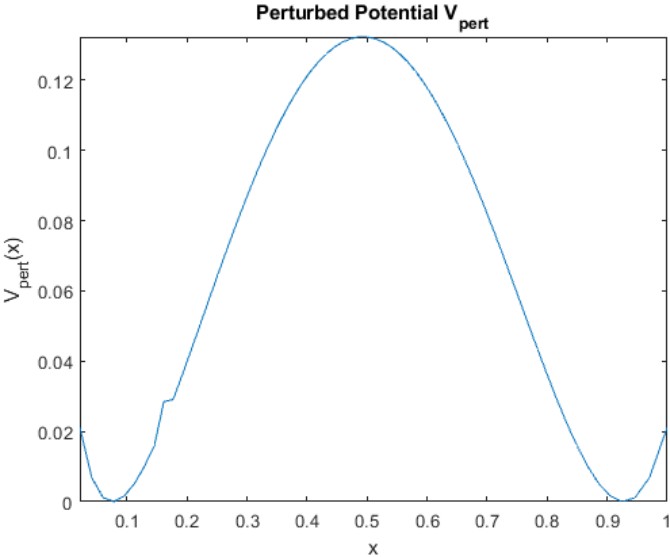

Figure 11: *Perturbed potential computed from the tridiagonal matrix $h_N^{CW} + \Delta V_N$ in the symmetric basis for $N = 65$, $b = 9/N$, $c = 1/45$, $d = 0.4$, $J = 1$ and $B = 1/2$. This potential has a 'flea' on the left side of well due the perturbation $\Delta V_N$.*

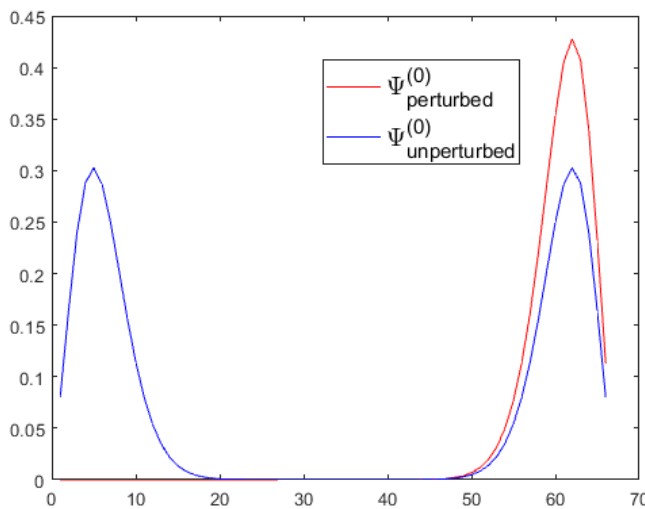

Figure 12: *Corresponding ground state (in red) of the perturbed Hamiltonian $h_N^{CW} + \Delta V_N$ is already localized for $N = 65$, $b = 9/N$, $c = 1/45$, $d = 0.4$, $J = 1$ and $B = 1/2$. Localization takes place on the right side of the well, since the flea raises the potential on the left side.*

## 5  Conclusion

We have established a link between the quantum Curie–Weiss Hamiltonian and a $1d$ Schrödinger operator describing a particle in a symmetric double well potential for $\hbar > 0$, where $\hbar = 1/N$. We have shown that the scaled quantum Curie–Weiss Hamiltonian restricted to the $(N+1)$-dimensional subspace $\mathrm{Sym}^N(\mathbb{C}^2)$ approximates a discretization matrix corresponding to this Schrödinger operator, defined on $L^2([0,1])$. Subsequently, we have shown that due to a small perturbation a $\mathbb{Z}_2$-symmetry of the Curie–Weiss model can already

be explicitly broken for finite $N$, resulting in a pure ground state in the classical limit. This confirms Anderson's mechanism for SSB in finite quantum systems (cf. the Introduction), but our specific "flea" perturbation came from similar results for Schrödinger operators with a symmetric double well potential in the classical limit $\hbar \to 0$ [1, 2, 26]. The results in these papers, which were obtained analytically, precisely match ours, obtained numerically, but this stil leaves the challenge of finding analytic proofs of our results. Furthermore, our approach, and especially the specific perturbations we use, should be extended to the case of continuous symmetries, where the relevant low-lying states (which for continuous symmetries are infinite in number as $N \to \infty$) now seem to be completely understood [4]. The dynamics of the transition from a localized ground state of the unperturbed Hamiltonian to a delocalized ground state of the perturbed Hamiltonian as $N \to \infty$ remains to be understood (this is an understatement); once achieved, it would perhaps also contribute to the solution of the measurement problem or Schrödinger Cat problem along similar lines [7, 8].

We finally discuss some of the correspondences as well as differences between the symmetry-breaking perturbations we used and those considered in the condensed matter physics literature. The key physical idea is the same in both cases:[18]

> "The general idea behind spontaneous symmetry breaking is easily formulated: as a collection of quantum particles becomes larger, the symmetry of the system as a whole becomes more unstable against small perturbations,"

where we add that the same is true as $\hbar$ becomes smaller at fixed system size, cf. [7] and references therein, and that the precise form of the instability is that for small $N$ or large $\hbar$ the perturbation plays almost no role (in either the spectral properties of the Hamiltonian or in its eigenfunctions), whereas for large $N$ or small $\hbar$ it metaphorically

> "can irritate the elephant enough so that it shifts its weight, i.e., we will see that the ground state, instead of being asymptotically in both wells, may reside asymptotically in only one well".

As explained in the Introduction, this accounts for the fact that real materials (which are described by the quantum theory of finite systems) do display SSB, even though the theory seems to forbid this. As we (and most condensed matter physicists) see it, these perturbations should arise naturally and might correspond either to imperfections of the material or contributions to the Hamiltonian from the (otherwise ignored) environment.

Mathematically, though, there are some differences between our mathematical physics approach to SSB in finite quantum systems and the standard theoretical physics one.[19] These differences are perhaps best explained by starting with the very *definition* of SSB. High sensitivity to small perturbations in the relevant regime are common to both approaches (as they are to the classical theory of critical phenomena and phase transitions, and even to complexity theory, which is full of bifurcations and tipping points, cf. [39]. In the physics literature this sensitivity to small perturbations is usually taken into account by adding an "infinitesimal" symmetry-breaking term like

$$\delta h_N = \varepsilon \sum_{x=1}^{N} \sigma_3(x) \tag{5.25}$$

to the Curie–Weiss Hamiltonian (2.1) and arguing that the correct order of the limits in question is $\lim_{\varepsilon \to 0} \lim_{N \to \infty}$, which gives SSB by one of the two pure ground states on the

---

[18]The first quotation is from Ref. [13] and the second one is from Ref. [2].

[19]See references in footnote 3.

limit algebra, the sign of $\varepsilon$ determining the direction of symmetry breaking. In contrast, the opposite order $\lim_{N\to\infty}\lim_{\varepsilon\to 0}$ gives a symmetric but mixed and hence unstable or unphysical ground state on the limit algebra.[20] Thus whenever there is difference

$$\lim_{\varepsilon\to 0}\lim_{N\to\infty} \neq \lim_{N\to\infty}\lim_{\varepsilon\to 0}, \tag{5.26}$$

this is taken to be a defining property SSB. This is valid, but it feeds the idea that, if SSB occurs, the limit $N \to \infty$ is "singular" (e.g. [13, 27, 28]), an idea that is increasingly challenged in the philosophical literature [9] and has almost disappeared from the mathematical physics literature.

Instead, we work with a definition of SSB that is standard in mathematical physics and applies equally to finite and infinite systems (provided these are described correctly), and to classical and quantum systems, namely that the ground state (suitably defined) of a system with $G$-invariant dynamics (where $G$ is some group, typically a discrete group or a Lie group) is *either* pure but not $G$-invariant, *or* $G$-invariant but mixed.[21] Accordingly, what is singular about the thermodynamic limit of systems with SSB is the fact that the exact *pure* ground state of a finite quantum system converges to a *mixed* state on the limit system.[22] However, this singular behaviour is exactly what is *avoided* by Anderson's tower of states triggered by the right perturbations, where a single limit (i.e. either $\hbar \to 0$ or $N \to \infty$) suffices,[23] in which the (still) *pure* ground state of the perturbed Hamiltonian (which is a symmetry-breaking linear combination of low-lying states) converges to some symmetry-breaking *pure* ground state on the limit system (be it a classical system or an infinite quantum system). The ensuing limit is then duly continuous in an appropriate meaning of the word (which it would *not* be without the perturbation mechanism).[24]

Of course, in order for this symmetry breaking to be *spontaneous* rather than *explicit*, the perturbation should be small to begin with, and should disappear in the pertinent limit.[25] As we have seen, it is easy to endow the "flea" $\delta V$ with these properties; for symmetry breaking in the double well potential all we need is that $\Delta E \to 0$ more rapidly than $\delta V \to 0$ as $\hbar \to 0$, cf. (4.22) - (4.24), and for symmetry breaking in the Curie–Weiss model the same holds for $N \to \infty$. Since in these models $\Delta E$ vanishes exponentially in $-1/\hbar$ or $-N$ as $\hbar \to 0$ or $N \to \infty$ (a fact about the spectrum that has nothing to do with the perturbations), this can hardly go wrong.

In this light, the following may help explain the relationship between our approach and the traditional one based on the non-commuting limits (5.26). For the latter, consider the $(x, y)$ plane with $x = 1/N$ (or $x = \hbar$) and $y = \varepsilon$, and for the argument in the previous paragraph,

---

[20]This procedure goes back at least to Bogoliubov [40], see also [41].

[21]See e.g. [7], Definition 10.3, page 379, and [42]. It may seem more natural to just require that the ground state fails to be $G$-invariant, but in the C*-algebraic formalism we rely on ground states that are not necessarily pure, which leaves the possibility of forming $G$-invariant mixtures of non-invariant states that lose the purity properties one expects physical ground states to have. Similarly for equilibrium states, where 'pure' is replaced by 'primary', which is a mathematical property of a pure thermodynamical phase. Order parameters *follow* from this definition, cf. §10.3, *loc. cit.*

[22]And similarly for the limit $\hbar$ of the symmetric double well system, where, as pointed out in the Introduction, the $\hbar = 0$ limit of the ground state is the mixed state (1.4), as opposed to, for example, a Dirac delta-function located between the wells (i.e. at $q = 0$). See also Refs. [2, 43]. This also explains the fact that the flea perturbations even work if their support is localized away from the bottoms of the two wells (or, equivalently, from the two peaks of the unperturbed ground state); see especially Ref. [26] for a detailed explanation of this point.

[23]This can also be achieved by letting $\varepsilon$ depend on $N$ in some suitable way, but if one does so one might as well drop the factorized form of perturbations like (5.25) altogether and admit more general expressions similar to the flea. Some literature indeed seems to do this, though not in a mathematically precise way.

[24]This is described via continuous fields of C*-algebras and states ( [7], Chapters 7–10).

[25]This must be taken to be a purely mathematical criterion, for real systems have $\hbar > 0$ and $N < \infty$, so that strictly speaking any form of symmetry breaking in Nature is explicit rather than spontaneous.

take $x = \delta V$ and $y = \Delta E$. In case of SSB, certain physical quantities $m(x, y)$ (like the magnetization) are discontinuous at $(0, 0)$ and hence their value at $(0, 0)$ depends on the path towards the origin. Eq. (5.26) expresses this path dependence in a crude way, which is captured by perturbations like (5.25), but the perturbations we consider follow specific *parametrized paths* towards $(0, 0)$. This explains why we are able to work with a single limit $\hbar \to 0$, since in our models $\Delta E = \Delta E(\hbar)$ is *given* (by the double well or Curie–Weiss Hamiltonian) and $\delta V = \delta V(\hbar)$ can be freely *chosen*, subject to the condition just mentioned, i.e. that the approach $\Delta E(\hbar) \to 0$ (as $\hbar \to 0$) must be quicker than $\delta V(\hbar)$.

The wealth of possible flea perturbations, as opposed to the more straightforward symmetry-breaking perturbations à la (5.25) considered in the condensed matter physics literature (i.e. coupling to a small constant external magnetic field) also weakens the critique expressed by Wallace [44], to the effect that cooling e.g. ferromagnets (but also antiferromagnets, (anti)ferroelectrics, ferroelastics, and superconductors) does not, experimentally, lead to a ground state in which the spins are all aligned with the external field, but rather to a state with various domains in which the spins are merely aligned locally but may differ quite randomly from domain to domain ( [45], pp. 346–354). Obtaining the right domain sizes admittedly requires fairly special fleas (as Wallace points out), but in the absence of any dynamical theory of cooling any argument in this direction, including this critique, is speculative. Wallace's suggestion that the formation of domains with specific sizes has already been explained by energetic considerations of the kind presented by Kittel (*loc. cit.*, attributed to Landau and Lifshitz), which even require the *absence* of constant external magnetic fields, is not effective against flea perturbations, which indeed are *required* to explain why *specific* domains in a *given* specimen (as opposed to *general* domains in *generic* materials) are formed; the problem is quite analogous to explaining SSB. Kittel also draws attention to the importance of singe-domain regions, e.g. for magnetic recording devices, but also in sedimentary rock formations.

Another objection to our approach to SSB, which equally well applies to the standard approach in condensed matter physics, is that a mechanism based on symmetry-breaking perturbations cannot be applied to gauge theories and hence cannot explain the Higgs mechanism. However, it is not local gauge symmetry that is broken in the Higgs mechanism, but a global symmetry, and by choosing gauge-invariant observables it is even possible to describe it without any reference to SSB, see Ref. [7], §10.10.

# A  Perron–Frobenius Theorem

In this appendix we provide the machinery for proving uniqueness and strict positivity of the ground state of the Curie–Weiss model for any finite $N$, based on the Perron-Frobenius Theorem. Though the result is well known, the precise combination of arguments is hard to find in the literature.

We start with some definitions and basis facts.

**Definition A.1.**    *1. A square matrix is called non-negative if all its entries are non-negative. It is called strictly positive if all its entries are strictly positive.*

*2. A non-negative matrix a is called irreducible if for every pair indices i and j there exists a natural number m such that $(a^m)_{ij}$ is not equal to zero. If the matrix is not irreducible, it is said to be reducible.*

*3. A directed graph is a graph $G = (V, E)$ with vertices V and edges E such that the vertices are connected by the edges, and where the edges have a direction. A directed graph is also called a digraph.*

698   *4. A digraph is called* *strongly connected* *if there is a directed path x to y between any two*
699   *vertices x, y.*

700   We use the notion of the directed graph or digraph of a square $N$-dimensional matrix $a$,
701   denoted by $G(a)$. We say that the digraph of $a$ is the digraph with

$$V = \{1, 2, ..., N\},$$
$$E = \{(i, j) |\ a_{ij} \neq 0\}.$$

702   The following result links irreducibility of a non-negative matrix to strongly connectedness of
703   its corresponding digraph. The proof is easy and therefore omitted.

704   **Lemma A.2.** *A non-negative square matrix a is irreducible if and only if the digraph of a is*
705   *strongly connected.*

706   We now come to the Perron-Frobenius Theorem. There are two versions of this theorem: one
707   for *strictly positive* matrices, and the other for *irreducible* matrices. We use the version for
708   irreducible matrices since the Curie–Weiss Hamiltonian $-h_N^{CW}$, represented with respect to the
709   standard basis for $\bigotimes_{n=1}^{N} \mathbb{C}^2$, is a non-negative and irreducible matrix of dimension $2^N$, as we
710   will see below.

711   **Theorem A.3.** *Let a be an $N \times N$ real-valued non-negative matrix, and denote its spectral radius*
712   *by $r(a) = \lambda$. If a is irreducible, then $\lambda = r(a)$ is an eigenvalue of a, which is positive, simple,*
713   *and corresponds to a strictly positive eigenvector.*

714   This theorem is based on properties of a matrix relative to some basis, so that the
715   Perron-Frobenius Theorem is valid if there exists a basis such that the matrix representation
716   of the operator in this basis satisfies the assumptions of the theorem. Note that multiplying
717   $-h_N^{CW}$ by $-1$, the eigenvalues will change sign and we find instead that the smallest eigenvalue
718   (i.e. the ground state) of $h_N^{CW}$ is simple and corresponds to a strictly positive eigenvector. As a
719   case in point, we are now going to prove a statement about our Hamiltonian $-h_N^{CW}$, relative to
720   the standard basis of $\mathbb{C}^2$ extended to a basis of the tensor product $\bigotimes_{n=1}^{N} \mathbb{C}^2$ in the usual way.

721   **Theorem A.4.** *The Curie–Weiss Hamiltonian $-h_N^{CW}$ from* (1.6), *represented in the standard basis*
722   *for $\bigotimes_{n=1}^{N} \mathbb{C}^2$, is non-negative and irreducible.*

723   *Proof.* Since all constant factors in $-h_N^{CW}$ are strictly positive, we only have to consider both
724   terms containing sums. We show that

$$\sum_{x,y \in \Lambda_N} \sigma_3(x)\sigma_3(y) \text{ and } \sum_{x \in \Lambda_N} \sigma_1(x) \tag{A.1}$$

725   are non-negative.     We have seen in the proof of Theorem 2.1 that the operator
726   $\sum_{x,y \in \Lambda_N} \sigma_3(x)\sigma_3(y)$ is a diagonal matrix with respect to the standard basis
727   $\{e_{n_1} \otimes ... \otimes e_{n_N}\}_{n_1=1,...,n_N=1}^{2}$ for $\bigotimes_{n=1}^{N} \mathbb{C}^2$. For non-negativity we must prove, independently of
728   the basis vectors, that there are at least as many plus signs as there are minus signs, i.e., we
729   have to show that

$$N^2 - 2n_+(N - n_+) \geq 2n_+(N - n_+). \tag{A.2}$$

730   This gives $N^2 - 4n_+(N - n_+) \geq 0$ if and only if $N^2 - 4Nn_+ + 4n_+^2 \geq 0$. The parabola

$$n_+ \mapsto N^2 - 4n_+N + 4n_+^2 \tag{A.3}$$

attains its minimum in $n_+ = N/2$, which is given by $N^2 - 4\frac{N}{2}n + 4(\frac{N}{2})^2 = 0$. So indeed, there are at least as many plus signs as minus signs, so that the corresponding diagonal term is non-negative. The other term $\sum_{x \in \Lambda_N} \sigma_1(x)$ does not contain any negative entries at all, so if we apply this to any basis vector $\{e_{n_1} \otimes \dots \otimes e_{n_N}\}$, we get a non-negative matrix. It follows that both operators in (A.1) are non-negative in the basis under consideration.

Now we show that the matrix corresponding to the Curie–Weiss Hamiltonian is irreducible. Note that irreducibility of a matrix does not depend on the basis in which the operator is represented, since similar matrices define equivalent representations which preserve irreducibility. We use Lemma A.2 to show that there is a direct path between any two vertices. But this is obvious: the operator $\sum_x \sigma_1(x)$ flips the spins one by one, and therefore the associated digraph is clearly strongly connected as we can find a directed path between any two vertices.[26] $\qquad\square$

By the Perron-Frobenius Theorem, the largest eigenvalue of the Curie–Weiss Hamiltonian $-h_N^{CW}$ is positive, simple and corresponds to a strictly positive eigenvector. This in turn implies that that the ground state eigenvalue of $h_N^{CW}$ is positive, simple, and has a strictly positive eigenvector.

# B  Discretization

This information provided in this appendix is based on Refs. [46–49]. These results are used above in §3.1. Recall from calculus that the following approximations are valid for the derivative of single-variable functions $f(x)$. The first one is called the *forward difference approximation* and is an expression of the form

$$f'(x) = \frac{f(x+h) - f(x)}{h} + O(h) \quad (h > 0). \tag{B.1}$$

The *backward difference approximation* is of the form

$$f'(x) = \frac{f(x) - f(x-h)}{h} + O(h) \quad (h > 0). \tag{B.2}$$

Furthermore, the *central difference approximation* is

$$f'(x) = \frac{f(x+h) - f(x-h)}{2h} + O(h^2) \quad (h > 0). \tag{B.3}$$

The approximations are obtained by neglecting the error terms indicated by the O-notation. These formulas can be derived from a Taylor series expansion around $x$,

$$f(x+h) = f(x) + hf'(x) + \frac{h^2}{2}f''(x) + \dots = \sum_{n=0}^{\infty} \frac{h^n}{n!} f^{(n)}(x), \tag{B.4}$$

and

$$f(x+h) = f(x) - hf'(x) + \frac{h^2}{2}f''(x) + \dots = \sum_{n=0}^{\infty} (-1)^n \frac{h^n}{n!} f^{(n)}(x), \tag{B.5}$$

where $f^{(n)}$ is the $n^{\text{th}}$ order derivative of $f$. Subtracting $f(x)$ from both sides of the above two equations and dividing by $h$ respectively $-h$ leads to he forward difference respectively the

---

[26]A different proof is given in Ref. [34], §5.3, p.78.

backward difference. The central difference is obtained by subtracting equation (B.5) from equation (B.4) and then dividing by $2h$.

The question is how small $h$ has to be in order for the algebraic difference $\frac{f(x+h)-f(x)}{h}$ (i.e. in this case the forward difference approximation) to be good approximation of the derivative. It is clear from the above formulas that the error for the central difference formula is $O(h^2)$. Thus, central differences are significantly better than forward and backward differences.

Higher order derivatives can be approximated using the Taylor series about the value $x$

$$f(x+2h) = \sum_{n=0}^{\infty} \frac{(2h)^n}{n!} f^{(n)}(x) \tag{B.6}$$

and

$$f(x-2h) = \sum_{n=0}^{\infty} (-1)^n \frac{(2h)^n}{n!} f^{(n)}(x). \tag{B.7}$$

A forward difference approximation to $f''(x)$ is then

$$\frac{f(x+2h) - 2f(x+h) + f(x)}{h^2} + O(h), \tag{B.8}$$

and a centered difference approximation is for example

$$\frac{f(x+h) - 2f(x) + f(x+h)}{h^2} + O(h^2). \tag{B.9}$$

Now we discretize the kinetic and potential energy operator. For simplicity, consider the one-dimensional case. We first discretize the interval $[0,1]$ using a uniform grid of $N$ points $x_i = ih, h = \frac{1}{N}, i = 0, 1, ..., N$. It follows that $f(x) \mapsto f(x_i) =: f_i$. The Taylor series expansion of a function about a point $x_i$ becomes

$$f_{i+k} = f_i + \sum_{n=0}^{\infty} (-1)^n \frac{(kh)^n}{n!} f^{(n)}(x), \tag{B.10}$$

where $k = \pm 1, \pm 2, ..., \pm N$. Similar as above, we can find central difference formulas for $f_j'$, $f_j''$, namely

$$f_j' = \frac{-f_{j-1} + f_{j+1}}{2h} + O(h^2) \tag{B.11}$$

$$f_j'' = \frac{f_{j-1} - 2f_j + f_{j+1}}{h^2} + O(h^2). \tag{B.12}$$

The approximations are again obtained by neglecting the error terms.

Using this uniform grid with grid spacing $h = 1/N$, it follows that the second derivative operator in one dimension is given by the tridiagonal matrix $\frac{1}{h^2}[\cdots 1 -2\ 1 \cdots]_N$ and the potential which acts as multiplication, is given by a diagonal matrix. With the notation $\frac{1}{h^2}[\cdots 1 -2\ 1 \cdots]_N$, we mean the $N$-dimensional matrix

$$\frac{1}{h^2} \begin{pmatrix} -2 & 1 & & & \\ 1 & -2 & 1 & \mathbf{0} & \\ & \ddots & \ddots & \ddots & \\ \mathbf{0} & 1 & -2 & 1 \\ & & 1 & -2 \end{pmatrix}.$$

Now suppose that the values of the kinetic energy operator $T$ are non-uniformly dependent of the positions in space. Then one needs to use a non-uniform grid in order to get a good description of the second derivative. We use the central difference approximation and approach $f$ by a Taylor series.

Denote $x_j$ by the $j^{th}$ grid point and $f_k = f(x_k)$. Then the Taylor series of $f$ at $x_j$ can be written as

$$f_k = f_j + \sum_{m=1}^{\infty} \frac{(x_k - x_j)^m}{m!} f_j^{(m)}. \tag{B.13}$$

If we let $h_j = x_{j+1} - x_j$, then similarly as above, for a three-point finite-difference formula i.e., only $f_{i+1}, f_i, f_{i-1}$ are used, we find that

$$f_{j+1} = f_j + h_j f_j' + \frac{h_j^2}{2} f_j'' + \frac{h_j^3}{6} f_j^{(3)} + \dots \tag{B.14}$$

and similarly one can write $x_{j-1} = x_j - h_{j-1}$, so that we find

$$f_{j-1} = f_j - h_{j-1} f_j' + \frac{h_{j-1}^2}{2} f_j'' - \frac{h_{j-1}^3}{6} f_j^{(3)} + \dots \tag{B.15}$$

Both expressions can be used to eliminate $f_j'$ to derive an expression for the second derivative:

$$f_j'' = \frac{2 f_{j-1}}{h_{j-1}(h_{j-1} + h_j)} - \frac{2 f_j}{h_{j-1} h_j} + \frac{2 f_{j+1}}{h_j(h_{j-1} + h_j)} + \frac{h_j - h_{j-1}}{3} f_j^{(3)} + \mathcal{O}(h^2). \tag{B.16}$$

This is the central difference approximation for the non-uniform grid. If we assume that $h_j - h_{j-1}$ is small, we may neglect the last term, and we get precisely that

$$\frac{2}{h_{j-1}(h_{j-1} + h_j)} = T_{j,j-1}, \tag{B.17}$$

$$\frac{-2}{h_{j-1} h_j} = T_{j,j}, \tag{B.18}$$

$$\frac{2}{h_j(h_{j-1} + h_j)} = T_{j,j+1}. \tag{B.19}$$

Therefore we find that the ratio, say $\rho_j$, equals

$$\rho_j = \frac{T_{j,j+1}}{T_{j,j-1}} = \frac{h_{j-1}}{h_j}. \tag{B.20}$$

Thus

$$h_{j-1} = \rho_j h_j. \tag{B.21}$$

We derive from this combined with the above three equations that

$$h_j^2 = \frac{2}{T_{j,j-1}\rho_j(1 + \rho_j)}, \tag{B.22}$$

$$\text{or} \quad h_j^2 = \frac{2}{T_{j,j+1}(1 + \rho_j)}. \tag{B.23}$$

## Acknowledgements

Chris van de Ven is Marie Skłodowska-Curie fellow of the Istituto Nazionale di Alta Matematica and is funded by the INdaM Doctoral Programme in Mathematics and/or Applications co-funded by Marie Skłodowska-Curie Actions, INdAM-DP-COFUND-2015, grant number 713485. Robin Reuvers is supported by the Royal Society through a Newton International Fellowship, and by Darwin College (Cambridge) through a Schlumberger Research Fellowship. The authors would like to thank both the referee (Jasper van Wezel) and Valter Moretti for their feedback, which has led to substantial improvements. We are also indebted to Aron Beekman for references on SSB in finite quantum systems.

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
