# Peer review of "Quantum spin systems versus Schroedinger operators: A case study in spontaneous symmetry breaking"

_SciPost Physics, doi:SciPost Phys. 8, 022 (2020)_

## Round 1 · Referee Report · Jasper van Wezel (Referee 1) · 2019-1-22

Strengths

1 - The submission has a clear and pedagogical style of presentation.
2 - All presented work is based on mathematically rigorous arguments.

Weaknesses

1 - The submitted manuscript contains only already-known physics
2 - It is not clear how the presented numerical results can be taken to the thermodynamic limit. This causes at least one serious issue in the manuscript (see report).

Report

This is a very nicely written paper discussing the emergence of spontaneous symmetry breaking in a version of the Curie-Weiss model. It includes a thorough treatment of many aspects of spontaneous symmetry breaking, in a mathematically rigorous fashion, which will be useful to any readers interested in the details of spontaneous symmetry breaking in general, and in the Curie-Weiss model in particular.

Beyond being a nice, clear, and precise description of the effects of a symmetry breaking field for finite N and hbar in a setting where it has -to the best of my knowledge- not been discussed in these terms before, the paper unfortunately does not offer much new insight. The idea that a finite symmetry-breaking field is required to break the symmetry of a finite system is completely standard. Similarly, the realisation that the strength of this field may be taken to infinitesimal values in the thermodynamic limit has been at heart of the way we teach spontaneous symmetry breaking for a long time. I appreciate the present submission as an explicit and pedagogical example of spontaneous symmetry breaking. Examples that can be treated in an accessible, yet mathematically rigorous, way are rare, and it is valuable to have more such examples. Beyond that, I don't see any new physics in this manscript. I would therefore urge the authors to rewrite their introduction in such a way that they make clear that this manuscript presents a new and pedagogical example of known physics, rather than a novel research result.

Besides this general remarks, I also noticed some other issues in the manuscript:

Major issues:

1- In the introduction, and in various places throughout the manuscript, the author fail to acknowledge, and even misrepresent, past results in the field. The realisation that finite symmetry breaking fields are needed to break the symmetry of finite systems has been around at least since the works of PW Anderson. The effect of finite perturbations and the need for suitable convergence in the thermodynamic limit has been discussed in detail for magnetic systems by various authors throughout the 1980's, and has been generalised to many other types of systems since. The authors fail to cite much of the previous work in this direction, and instead present their own papers from the past few years as if these are the first results in the field. 1b - The failure to acknowledge previous work is made worse by the fact that the authors use their own personal terminology for concepts that have been discussed in the literature for many years under other names. For example, the "flea in Schrodinger's cat" is nothing other than the "symmetry breaking field" in any other discussion of spontaneous symmetry breaking.

2 -Although I agree with the idea of the discussion in section 4.1, I am not convinced the perturbation shown in figure 9 can be regarded as a proper choice for a symmetry breaking field. As the authors show in their manuscript, the two nearly degenerate ground states become arbitrarily well-localised in the limit of large N. Taking the limit of N to infinity while keeping the perturbation finite thus means that the ground state wavefunction will have zero overlap with any alterations of the potential away from the minima of the double well. Only perturbations that make the two minima in the potential non-degenerate can then serve as proper symmetry-breaking fields in the limit of large N. In other words: I suspect that for the perturbation shown in figure 9, the energy difference between the nearly degenerate ground states scales as lambda/N, implying that N can always be chosen large enough for the perturbation chosen by the authors to be practically irrelevant. This is precisely opposite to the effect of a proper symmetry breaking field, which causes an energy splitting proportional to lambda*N, and which becomes more relevant with increasing N.

Minor issues:

1 - On the bottom of page 9, and again in various places further on, it is confusing to talk about "new ground states". Two states simply become degenerate up to numerical precision, and any superposition of them is a (numerically) allowed ground state. If you always find a particular "new" ground state in the numerics, such as a localised state, that must mean there is a symmetry breaking field in your numerical implementation of the calculation. Without such a field, you ought to find random superpositions of the two degenerate states. This should really be explained already on page 9. On a related note, the number 80 in N=80 is arbitrary, and must depend on the machine precision of the computer used to do the calculations. The authors really ought to mention that.

2 - in the first line of the introduction, the authors choose ferromagnets as their example for the archetype of spontaneous symmetry breaking. I suspect the authors are aware that this is actually a very bad example. Ferromagnets are very special in the sense that their symmetry-broken state is an exact eigenstate of their symmetric Hamiltonian for any N. This is not at all generic, and in fact severely affects the physics associated with symmetry breaking in ferromagnets (see for example 10.1016/j.aop.2015.07.008 and references therein).

3 - It seems like the matrix elements discussed in theorem 2.1 are included in those of appendix 2 in 10.1103/PhysRevB.74.094430 ?

Requested changes

1 - I suggest the authors rewrite their introduction in such a way that they make clear that this manuscript presents a new and pedagogical example of known physics, rather than a novel research result.

2 - Please add references to and discussion of previous results in the literature.
I would also suggest using, or at least mentioning, standard terminology.

3 - The authors should discuss the behaviour of their symmetry broken states with increasing N, in order to establish a connection between their low-N numerics, and the large-N limit that is relevant for actual physical systems.

4 - Please explain clearly that finding particular "numerical ground states" is the effect of choosing a particular numerical implementation for the calculation, which is equivalent to including a symmetry breaking field.

  • validity: high
  • significance: ok
  • originality: ok
  • clarity: high
  • formatting: good
  • grammar: excellent

Author:  Klaas Landsman  on 2019-02-01  [id 419]

(in reply to Report 1 by Jasper van Wezel on 2019-01-22)

Dear referee (Jasper) and editor, We are very grateful for this detailed and elaborate report and enter this correspondence trusting we will reach an agreement. The correct literature attributions, though important, are secondary and will be discussed below. Our primary point, in respectful disagreement with the referee, concerns the novelty of the paper. The specific mechanism we use (and sometimes may even appear to propose) for SSB was never meant to be new; anything in this direction was meant to be expository and we are pleased that the referee has appreciated this expository aspect.

Though we acknowledge, with hindsight, that both the abstract and the order of presentation in the Introduction are evidently misleading in this respect, the main novel point of the paper (in section 3) is the discovery (as we believe it to be) that the ground state of the quantum Curie-Weiss model (in a small magnetic field), compressed onto the N+1 dimensional subspace in which (due to permutation symmetry) the ground state must lie (as explained in section 2), is the same as the ground state of a discretized double well potential on the line. For this we provide both extensive numerical and analytic evidence. This direct link between a quantum spin model and a discretized Schroedinger operator came to us unexpectedly, and we were unable to find anything like it in the literature: what has been remarked so far were (largely spectral) analogies between say the quantum Ising model and continuum double well potentials; what we provide is a direct mapping to the discretized double well potential, with splendid convergence properties for large N. So we ask the referee to either acknowledge this novelty or provide literature where this discovery is predated or at least anticipated (of course, we have done our best to find such papers ourselves but were unable to - we have obviously been unable to familiarize ourselves all the thousands of papers on these topics).

A second novel point is the specific perturbation we propose in section 4 for the analogue of what we call the "flea" perturbation for the continuum double well potential (see below for terminology and priority issues etc.) for the Curie-Weiss model. This was actually quite hard to find, and our analysis shows that and how it does the job. In response to point 2 by the referee, despite its decreasing behaviour for large N (which is needed since the energy gap between the original ground state and the perturbed ground state must of course go to zero for N to infinity), according to which it vanishes as an inverse power of N, it is still big enough to tilt the ground state into a symmetry-breaking one, and this is, just like the double well case, because, small as it is, the perturbation still dominates the energy gap in the unperturbed model. This brings us to the third point:

  1. The question how well known the mechanisms for SSB we use is, and who should be credited and cited for it. We feel there is a genuine mismatch here between the theoretical and mathematical physics literature. Our starting point was the latter, in which the paper [10] by Jona-Lasinio (also a famous theoretical physicist, by the way) was seen as a sensation at the time, which triggered a flurry of literature that more or less ended with ref. [19] by Simon, following which the case of the symmetric double well was regarded as completely understood (at least statically - the dynamical transition is still unclear, as we agreed in recent conversation in response to a paper by Wallace). The "flea" terminology is quite common in mathematical physics for denoting this mechanisms for symmetry breaking; Simon called it the "Flea on the elephant", and we, in later work related to the measurement problem, renamed it the "flea in Schroedinger's Cat", a light-hearted name that has been picked up in the foundations of physics literature in which we introduced this technique. We then started looking for a similar mechanism for quantum spin systems, found the 1994 paper by Koma and Tasaki (ref. [10]) which contains everything expect the specific perturbation, and later the papers by the referee (some with Zaanen and some with Van den Brink) which contain the perturbation but not the analysis of Koma and Tasaki; we have simply combined the two. We cannot speak on behalf of the theoretical physics/condensed matter physics community, but trust the referee that this mechanism has been well know for many decades, but we would be interested in giving specific references, since we were unable to find these ourselves; at least standard textbooks (both on condensed matter and on high-energy physics) do not explain this mechanism or even come close to it. Again, the point we mean is that SSB originates in the interplay between the low-lying states and their response to (almost) arbitrarily small symmetry-breaking perturbations: no textbook we are familiar with mentions this, and papers by Anderson at best give some sort of intuition in this direction falling short of anything like a detailed mathematical analysis. We would really be grateful for specific (and earlier) references beyond those to Van Wezel et al we already give. Again, this has nothing to do with what we claim to be new in our paper; it would just improve the exposition and give credit where credit is due.

Further, smaller details a matters of slight disagreement can be settled in the next stage, without any doubt. Let us again express our gratitude to the referee and the editor for their efforts spent on our paper so far and in the future.

Jasper van Wezel  on 2019-02-06  [id 429]

(in reply to Klaas Landsman on 2019-02-01 [id 419])

Dear authors,

Thank you very much for your detailed reply, and for explaining the background to your presentation. I would like to answer the points made in your reply:

  • Although I appreciate the effort that the authors put into constructing the mapping between the Curie-Weiss model and the discretised double well system, I'm afraid I don't share their surprise at finding such an equivalence. Let me stress that the mapping is not exact for general N. Rather, I believe the authors show that the large-N limit of the discretised double well model can be mapped onto the Curie-Weiss problem. But all this means, is that for large enough N the total magnetisation may be considered to be of fixed amplitude, while the potential determining its direction has a discrete set of minima which, again for large enough N, can be approximated to be locally harmonic. All of these seem to me to be generic properties of interacting spin systems in which the symmetry is reduced to be discrete due to an applied field.

Please allow me to be clear: I do acknowledge that the specific example of correspondence between the two models found by the authors in the present work may not have been known before (I have not encountered it, as far as I'm aware), and may be new in that regard. So if the authors insist that the primary novelty of their work is in the mapping between models and in the choice of symmetry breaking field, and that the rest is primarily educational, I accept that. I would just like to point out that the novelty claimed by the authors seems limited if seen in the light of the many related results already in the literature, and I would ask the authors to revise their paper to acknowledge that. I would suggest that instead, they can emphasize the important educational aspect of their work.

  • I appreciate the point made by the authors that there may be a mismatch in the extent to which spontaneous symmetry breaking has been covered in the condensed matter versus mathematical physics literature. However, since the authors address a typical condensed matter topic (that of order in spin systems), I think they should respect the established condensed matter physics literature. In terms of standard terminology, this means they should at least mention commonly used and well-established names like "symmetry breaking field" in favour of "flea". In terms of giving credit to work in the literature on this subject, I would like to at least mention the work of Lieb and Mattis ( J. Math. Phys. 3, 749 (1962)) showing how energy levels are ordered in magnetic systems in general, and that the exact ground state of any magnetic system (barring the pure ferromagnet) is symmetric; the works of both Kaiser and Peschel (J. Phys. A 22, 4257 (1989)) and of Kaplan, Von der Linden, and Horsch (Phys. Rev. B 42, 4663 (1990)) which both discuss how the symmetry in the antiferromagnet is broken spontaneously, including the role of the symmetry breaking field as well as that of the spectrum of low-energy states; and finally, the work of Bernu, Lhuillier, and Pierre (Phys. Rev. Lett. 69, 2590 (1992)) giving a numerical treatment of symmetry breaking in finite antiferromagnets (particularly relevant with regard to the present submission). I believe these works together established the modern understanding of spontaneous symmetry breaking in magnetic systems. More recently, a lot of progress has been made by various authors, including Watanabe, Brauner, and Beekman, to name just a few. There are certainly many more relevant references that could be added: this is an active field of research, building on a long history. I do agree with the authors that it is not easy to find a textbook that explains the particular aspects of symmetry breaking which are the subject of the current paper in a clear and concise manner. I can however point the authors to a set of lecture notes written in 1994 which shows that, even then, the subject was already covered in detail in (good) condensed matter lectures at the MSc level. Finally, although cryptic in some aspects, the papers of Anderson were certainly instrumental in establishing and promoting this modern understanding of spontaneous symmetry breaking in condensed matter theory.

  • Finally, with regard to their choice of symmetry-breaking term, I don't think the authors address my concern at all. As I wrote before: " I suspect that [..] the energy difference between the nearly degenerate ground states scales as lambda/N [..] This is precisely opposite to the effect of a proper symmetry breaking field, which causes an energy splitting proportional to lambda*N, and which becomes more relevant with increasing N." In their reply, the authors simply state : "despite its [.. vanishing ..] as an inverse power of N, it is still big enough to tilt the ground state into a symmetry-breaking one." But this is precisely the thing I question. I do not think that it is enough to tilt the ground state into a symmetry-broken one. My complaint is precisely that from the calculations shown in the current manuscript, it is impossible to tell whether or not the symmetry will be broken in the limit of large N (rather than for N<80). Based on the fact that the wavefunction localises exponentially in the centre of the well, and thus has an exponentially small overlap with the perturbation chosen by the authors, one would expect the influence of the perturbation to disappear in the limit of large N. If the authors want to argue that their perturbation is nonetheless a suitable symmetry-breaking field, they will have to show explicitly that how the symmetry-breaking effect scales with system size, or equivalently, that the limits of large N and vanishing lambda do not commute. On a related note: the authors mention they found choosing a suitable symmetry-breaking field to be "quite hard". I do not really understand this, since generically, if a symmetry is to be broken by any infinitesimal perturbing field, the perturbation should be conjugate to the broken symmetry. In this case, that simply means that any suitable symmetry breaking field raises the (minimum) energy of one of the potential wells with respect to the other. The fact that the perturbation chosen by the authors does not do this, is another reason for suspecting that their choice of symmetry-breaking field may not be adequate in the large-N limit.

  • The authors did not yet address any of the other points raised in my report, which therefore remain open.

---

## Round 2 · Referee Report · Jasper van Wezel (Referee 1) · 2019-6-26

Strengths

1 - The submission has a clear and pedagogical style of presentation.
2 - The presented mathematical mapping between models is rigorous.

Weaknesses

1 - Besides introducing a mapping between the Curie-Weiss model and a Schrödinger operator, the submitted manuscript does not contain any new physics.
2 - Two of the main claims of novelty are simply false (see below).

Report

Dear authors,

thank you very much for seriously considering the suggestions and comments made in my previous report. I believe the resubmitted manuscript has a much improved discussion of how the present work relates to previous literature, and a much more balanced presentation of the current results.

I still think the discussion in the present paper is a worthwhile pedagogical example of spontaneous symmetry breaking in an accessible, yet mathematically rigorous, setting. I also appreciate the point of the authors that the presented mapping from the Curie-Weiss model onto a discretized 1d Schroedinger operator with double well potential is new.

However, I also continue to be of the opinion that the other two points presented by the authors as their main results, are not new:

1) In the final page of the conclusions, the authors emphasize they believe one of the main differences between their work and the standard approach to SSB in the cond-mat literature, is that their approach involves only a single limit, rather than two non-commuting ones.

In fact, this is simply untrue. The authors do use two limits, and they do not commute.

The authors even write this themselves, in the conclusions, where they state: "all we need is that ∆E → 0 more rapidly than δV → 0 as hbar → 0". The two limits taken by the authors are ∆E → 0 (which corresponds to N → infinity), and δV → 0. These are precisely the same two limits as the ones in equation (5.26), used in the standard SSB literature. In both cases, they do not commute. The trick played by the authors, of parameterising both ∆E and δV as functions of hbar, and then requiring that ∆E vanishes more rapidly than δV, is just precisely equivalent to taking the limit N-> infinity before taking the limit δV → 0.

1b) The same point comes up in a second guise on page 5, where the authors write: "on a first analysis (to be corrected in what follows!) there is no ssb for any finite N"

The authors seem to suggest here that there is SSB for finite N. If so, then for which N? The only possible answer is some scale set by the ratio of the two energy scales in the model: that of the double well potential, and that of the "flea". In other words, for symmetry breaking in finite N, a non-zero "flea" is required. Spontaneous symmetry breaking (with an infinitely weak "flea") is possible only in the infinite N limit. So again, we see that two non-commuting, singular limits are in fact present in the author's work, and they appear in precisely the same way as in the standard approach.

2) In section 4, the authors argue there is a fundamental difference between their "flea" potential, and the symmetry-breaking perturbation introduced in the standard SSB literature.

Again, this is simply not true.

The requirements on the "flea" potential formulated between equations 4.2 and 4.3 are precisely such that the support of the "flea" coincides with the tails of the gaussian (localised) ground states. Of course this must be the case, for if the support of the flea would not coincide with the support of either ground state, it could have no effect.

As the authors argue in several places in the article, the width of the Gaussian ground state in the scaled coordinates is proportional to 1/sqrt(N), so that the ground states become exponentially localised in the middle of the wells in the limit N-> infinity. In that limit then, the "flea" must have support in the middle of the well, and corresponds simply to a lifting of the degeneracy between the two wells, which would be the standard symmetry-breaking field proposed by the standard cond-mat SSB literature.

It should be noted here that the authors seem to suggest on page 21 that the "flea" is strictly zero in the centre of the well. This cannot work in the limit of infinite N. Again, the argument is simple, and was already made in my first report: in the infinite N limit, the ground state is a Dirac delta distribution at the well centre. If the perturbation is zero there, the two states localised in either of the two wells are strictly degenerate, and a single symmetric superposition of them will be the unique ground state.

The authors can probably not see this in their numerical approach, since it cannot go beyond N=60, but the fact that there cannot be an effect of any "flea" that is zero in the centre of the wells in the infinite N limit is obvious.

Besides these major points, I also noticed some minor things:

  • on page 20, the authors mention several energy scales that do not have the units of energy. I suppose they mean these energies are for certain specific values of B and J? Even so, they really ought to write them in terms of B and J, or introduce some units.

  • The result of equation 3.36 simply shows that the double well potential is locally quadratic and therefore has the spectrum of a harmonic oscillator. It would be good to point this out, and also to point out that the approximation will break down at higher excitation energies.

In conclusion, I remain of the opinion that the discussion provided by the authors is worthy of publication, as a mainly education piece with perhaps the slight novelty of formulating a new mapping between the Curie Weiss model and a Schroedinger operator.

I do not believe, however, that the specific limit and the specific form of the perturbation considered by the authors are different from those used in the standard SSB literature, and I therefore do not judge this paper to be a significant advancement of the field.

---

## Round 2 · Author Response

We are very grateful for the time and energy spent by the referee on this paper, both through SciPost and through personal correspondence. The paper is much more balanced now and has a considerably improved bibliography. As to the novelty of the paper as opposed to its review character, we have now made it clearer where we feel out contributions lie. Mapping the Curie-Weiss model onto a discretized 1d Schroedinger operator with double well potential is one of those, based on a reduction of the model due to permutation symmetry as explained in section 2 (while such mappings in other models were known, and are referred to, this case seems to be new). The other, we continue to maintain, is the specific mathematical form of the symmetry-breaking perturbations of the CW model. It should be clear from this revision that we do not claim any physical novelty, but the mathematics is, as far as we know, even given the many new references added thanks to the referee, still new, though, as we wrote from the beginning, directly inspired by work in the 1980s on the classical limit of Schroedinger operators. In order to make this clearer than in v1, we have added a lengthy discussion of our perturbation in section 4, which was lacking in v1, and we have also slightly modified the perturbation in order to make the analogy with the "flea" from the 1980s clearer, including redoing the numerical simulations, which gave almost exactly the same results as before. We have also made a comparison with the condensed matter physics literature on SSB (in so far as we know it) in section 5, which emphasises our different starting point for SSB (namely a state decomposition perspective as opposed to a non-commuting limits perspective). The Introduction has also been modified in order to give the right historical perspective (as we see it). In this way all major and minor points raised by the referee have been taken into account or answered.

---

## Round 2 · List of Changes

- slight rephrasing of abstract
- section 1: new footnote 3 including many new references (see below) and embedding of footnote in main text. New paragraph at the end added.
- section 2: small corrections only
- section 3: streamlined, all details on discretization procedures that we eventually decided not to use (but felt it worth explaining why not) now deleted
- section 4: 2 x 2 matrix illustration added, change in "flea" family of perturbations (physics the same as before, math now more congruent with the Schroedinger operator case, numerics redone with similar results).
- section 5: Text after first paragraph (which largely comes from v1) added.
- References: nos. 2, 4, 5, 6, 7, 8, 9, 10, 15, 19, 22, 23, 24, 31, 32, 33, 35, 36, 37, 40, 42, 48, 49 added (about half suggested by the referee and half found by ourselves).

---

## Round 3 · Referee Report · Jasper van Wezel · 2019-8-16

Report

I thank the authors for the added discussions.

The new paragraphs do clarify the difference between the CM and MathPhys literature on this topic, and thus alert the reader to look into this further. However, they do not add any arguments to the discussion between the author and myself, and I am not convinced that any of my previous objections were invalid.

I also do not have any new arguments myself, beyond what I already wrote in my previous report. I therefore recommend that the editor decides whether or not they are happy to publish, taking into account the differences of opinion and arguments in the already available reports.
* * *
To the authors: I would like to clarify one thing - that my earlier points 1b and 2 do not contradict each other.
1b points out that the approach of the "flea" advocated by the authors is the same as that of a "symmery-breaking field" which is standard in the CM literature.
2 points out that one does have to choose an appropriate symmetry-breaking field, and that the authors have not at all proven their perturbation to remain appropriate all the way to the thermodynamic limit.

  • validity: -
  • significance: -
  • originality: -
  • clarity: -
  • formatting: -
  • grammar: -

Author:  Klaas Landsman  on 2019-08-19  [id 582]

(in reply to Report 1 by Jasper van Wezel on 2019-08-16)
Category:
answer to question
reply to objection

Hi, We agree to now leave this to the editor and emphasize that we have followed the previous editorial recommendation to (re)submit a minor revision. On the one hand, the dialogue with the referee has improved the paper almost beyond recognition, but on the other hand we regret that no complete agreement has been reached. Apparently, the subject lends itself to various interpretations, like works of modern art. We do believe that we answered all points raised by the referee and our previous replies explain in considerable detail (or at least intended to explain in detail) how we did so.

In reply to the last two points made by the referee we would like to point out that: 1) Apparently no perturbation is regarded as novel (by the referee) as long as it falls within the scope of general symmetry-breaking terms. Of course our flea perturbation falls within that scope - the novelty (or so we think) lies in its detailed mathematical form. We are not P.W. Anderson and we emphasized that this novelty is limited (but in our view still very interesting) from the start. 2) It is true that our evidence for symmetry breaking triggered by our perturbation is numerical and of course these were carried out for finite N, and hence not "all the way to the thermodynamic limit". However, the totally analogous flea perturbations for the double well potential have been rigorously proven, in the papers we cite throughout ours, viz. refs. [20], [15], and [40], to do this job for hbar -> 0, and since our perturbation is obtained from that one through the mapping explained in chapter 3, under the substitution hbar -> 1/N, we in fact collect two kinds of evidence for its validity. Hence we take issue with the claim that we "have not at all proven [our] perturbation to remain appropriate all the way to the thermodynamic limit". Since our mapping is itself only defined for finite N, we would agree that we have not strictly proved this (i.e. analytically), but inserting "at all" wrongly suggests that we have no evidence at all (sic). See especially page 26.

---

## Round 3 · Referee Report · Bruno Nachtergaele · 2019-12-14

Report

Spontaneous symmetry breaking plays a central role in many physical theories and our observed physical world exhibits symmetry breaking in many situations. The purpose of this work is to help clarify a subtlety in the way spontaneous symmetry breaking manifests itself in statistical mechanics.

A quantum system with a finite number of degrees of freedom has a unique equilibrium state and, more often than not, the ground state is also unique. These states then necessarily share all the symmetries of the Hamiltonian. Spontaneous symmetry breaking manifests itself in the thermodynamic limit but, of course, the state obtained as a limit of the unique and symmetric finite-system states, will also be symmetric. These symmetric states are a uniformly weighted mixture of the distinct pure symmetry broken phases. In the physical world, under most conditions, only pure phases are observed. Therefore, it is generally accepted that the extremal states found in an ergodic decomposition (at positive temperatures) or into pure states (at zero temperature), are the ones that describe physical reality.

The standard procedures in statistical mechanics to construct the pure phases involve adding symmetry breaking terms to the Hamiltonian, such as a uniform magnetic field that favors a particular value of the magnetization in a ferromagnet, and letting the field strength vanish in the thermodynamic limit. Another approach is to use symmetry breaking boundary conditions that recede to infinity in the thermodynamic limit. These two commonly used approaches work well. For those who want to understand what happens in an experiment on a finite sample, however, these procedures are not satisfactory. Pure phases are observed without using a magnetic field or a particular boundary condition. The accepted explanation for this is that any realistic preparation process will inevitably exhibit an instability that steers the end result to one of the pure phases.

In this paper, the authors study aspects of this instability in a particular model and take inspiration from an analogy between the thermodynamic limit and the classical limit, which I now briefly explain.

The thermodynamic limit turns the order parameter characterizing a phase transition with spontaneous symmetry breaking into a classical observable, with a deterministic value labeling the particular pure phase the system is in. The authors note the following analogy with the emergence of spontaneous symmetry breaking in the classical limit of a quantum particle in a symmetric double-well potential. In that system, there are two energy minimizing states of the classical particle and only states where the particle is sitting in one of the minima are physical. At positive values of Planck's constant, the system has a unique ground state with the symmetry of the potential and this symmetry is preserved in the classical limit. In other words, the classical limit ground state is a mixed state with equal probability of finding the particle in either minimum.

In the 1980's, mathematical works appeared that showed how a vanishingly small perturbation of a double-well potential, the proverbial flea on an elephant (which does not need to break the degeneracy of the classical energy minima) leads to a classical limit where the particle occupies just one of the minima. In the present work the authors show an analogous behavior in the thermodynamic limit of the mean-field Ising model (aka the Curie-Weiss model) in a transverse external field. For small transverse field, the spin flip symmetry is spontaneously broken in the ground state. A specific class of tiny perturbations is introduced and it is shown with the aid of numerical calculations that they steer the ground state in the thermodynamic limit to one of the pure ground states. In the process, the analogy between the thermodynamic limit of the Curie-Weiss and the classical limit of a double well system is made concrete by a mathematical reduction of the first to an instance of the second.

The value of this work is primarily of an illustrative and pedagogical nature. The apparent contradiction between the standard story and the physically observations is often a stumbling block for teachers and students alike. This paper, as its title promises, presents a case study that validates the explanation based on a generic instability. I found the paper to be very well-written. The cited references certainly do not exhaust the literature on the topic but they do provide sufficient background, historic context, and pointers to interesting further reading.

I believe this work will be a useful addition to the literature that may help people think more clearly and in mathematically precise terms about the important physical phenomenon of spontaneous symmetry breaking. It is unlikely to be the last word on the topic, but I expect to be a widely appreciated contribution of lasting value.

I recommend that this paper be accepted for SciPost in its present form.

---

## Round 3 · Author Response

Since only a minor revision was asked, we tried to clarify remaining differences with the referee in a revised Discussion session, see List of Changes below. This fully answers point 1a) in the last referee report, which seems to largely concern a way of phrasing things rather than a fundamental difference of opinion, and in this new version we hope to do justice to both our own views and the referee's; see especially the new lines 9-19 on page 29. Points 1b) and 2) are more difficult to deal with, since the criticisms seem contradictory to us: on the one hand (1b) our perturbations are claimed to be the same as those in the theoretical condensed matter physics literature, while on the other hand (2) they can't work because they are localized off the bottoms of the wells, or, equivalently, away from the double peaks of the unperturbed ground state wave function. Point 2) applies verbatim to our key mathematical references (Jona-Lasionio et al, 1981; Graffi et al, 1984; Simon, 1985) which are well-attested and uncontroversial papers with a large follow-up in mathematical physics, so here we suggest that physical intuition should bow to mathematical rigor - in fact the referee's reaction (which if correct would invalidate this entire body of work, on which, as we explain in the unrevised Introduction, our entire approach is based) confirms how completely unexpected these papers were at the time, and in our view still are; our mission with this paper is partly to advertise this work, which so far has not landed at all in theoretical physics. We answer the particular technical point made by the referee in our new footnote 22 on page 28. In our view, this also dispels 1b), but we do not make a big song and dance about the novelty of the flea perturbations because, as we explain throughout the paper, we merely transfer them from the double well potential (in which context they were introduced in the literature just cited) to the Curie-Weiss model.
The minor points made by the referee are uncontroversial and have been incorporated. On the whole, our dialogue with the referee and the journal has greatly improved and clarified our paper so we repeat our gratitude, simultaneously hoping that the end has now been reached! In particular, we hope that the referee and the editor accept that there is not a single valid perspective on SSB; there is a genuine difference between the standard theoretical physics approach (which the referee follows) and the standard mathematical physics approach (which we follow), each with its own advantages, and this paper is partly meant as a bridge between the two. See also the list of changes below.

---

## Round 3 · List of Changes

page 17, last line added (answers minor point by referee)

page 20, sentence after (3.36) added, following referee almost verbatim

page 28, discussion after (5.26) rephrased to clarify that there are two different points of view on SSB, where we explain why we favour ours (which is the standard one in mathematical physics), upon which the entirely new paragraph in lines 9-19 on page 29 relates the two approaches again, in a way which seems to clarify the situation, leading to peaceful coexistence, or so we hope.

Reference 17 added (Note that a huge addition to the bibliography has already been made in version 2; this one was added to support our new footnote 22, see above)

---

## Editorial Decision

published